



# Recurrent Rossby waves during Southeast Australian heatwaves and links to quasi-resonant amplification and atmospheric blocks

S. Mubashshir Ali*[1], Matthias Röthlisberger[2], Tess Parker[3], Kai Kornhuber[4], and Olivia Martius[1,5]

1. Oeschger Centre for Climate Change Research and Institute of Geography, University of Bern, Bern, Switzerland,
2. Institute for Atmospheric and Climate Science, ETH Zürich, Zürich, Switzerland,
3. School of Earth, Atmosphere and Environment, Monash University, Clayton VIC, Australia,
4. Earth Institute, Columbia University, New York City, NY, USA, and
5. Mobiliar Lab for Natural Risks, University of Bern, Bern, Switzerland.

*Corresponding author: mubashshir.ali@giub.unibe.ch

**Abstract**

In the Northern Hemisphere, recurrence of transient Rossby wave packets over periods of days to weeks, termed RRWPs, may repeatedly create similar weather conditions. This recurrence leads to persistent surface anomalies and high-impact weather events. Here, we demonstrate the significance of RRWPs for persistent heatwaves in the Southern Hemisphere (SH). We investigate the relationship between RRWPs, atmospheric blocking, and amplified quasi-stationary Rossby waves with

two cases of heatwaves in Southeast Australia (SEA) in 2004 and 2009. This region has seen extraordinary heatwaves in recent years. We also investigate the importance of transient systems such as RRWPs and two other persistent dynamical drivers: atmospheric blocks and quasi-resonant amplification (QRA).

We further explore the link between RRWPs, blocks, and QRA in the SH using the ERA-I reanalysis dataset (1979–2018). We find that QRA and RRWPs are strongly associated: 40% of QRA days feature RRWPs, and QRA events are 13 times

more likely to occur with an RRWPs event than without it. Furthermore, days with QRA and RRWPs show high correlations in the composite mean fields of upper-level flows, indicating that both features have a similar hemispheric flow configuration. Blocking frequencies for QRA and RRWP conditions both increase over the south Pacific Ocean but differ substantially over parts of the south Atlantic and Indian Ocean.



## 1. **Introduction**

Since 1900, extreme heat has been responsible for more fatalities in Australia than all other natural hazards combined

(Coates et al., 2014). Heatwaves also exacerbate the risk of wildfires, cause surges in power demand, and increase insurance

costs (Hughes et al., 2020; Insurance Council of Australia, 2020). Increasingly frequent and severe heatwaves in the

midlatitudes in the recent years (Coumou et al., 2013; Perkins-Kirkpatrick and Lewis, 2020; IPCC 2021) have spurred

fruitful research on the atmospheric drivers of heatwaves. Understanding the dynamical mechanisms is particularly

important for improving sub-seasonal prediction (Quandt et al., 2017) and for quantifying future changes in heatwaves

(Wehrli et al., 2019; Shepherd, 2014). Several large-scale atmospheric mechanisms and phenomena have been identified as

potential drivers of heatwaves in the extra-tropics. These include blocking anticyclones (e.g., Barriopedro et al., 2011;

Drouard and Woollings, 2018), amplified quasi-stationary waves (Teng et al., 2016; Kornhuber et al., 2020), and recurrent

Rossby wave patterns (Röthlisberger et al., 2019). However, these phenomena have mainly been studied in isolation. Here,

we focus on these three large–scale dynamical drivers of heatwaves to explore their relative importance, co-occurrence, and

potential interactions during heatwaves in south-eastern Australia (SEA).





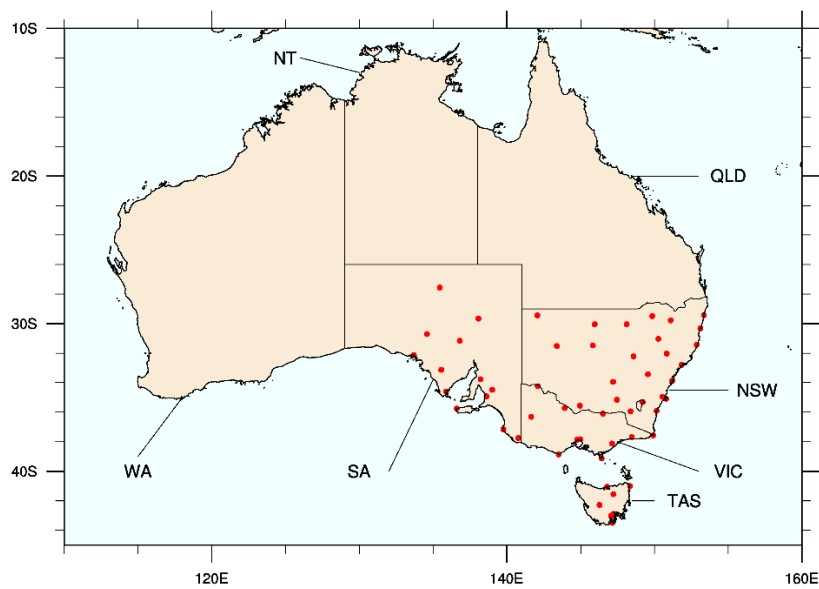

**Figure 1. Map of Australia showing the states of Southeast Australia (SEA): South Australia (SA), Tasmania (TAS), Victoria (VIC), and New South Wales (NSW). Other states shown are Queensland (QLD), Northern Territory (NT), and Western Australia (WA). Red dots indicate Australian Bureau of Meteorology's (BoM) monitoring stations used in this study (see Methods).**

Broadly, heatwaves in SEA (Fig. 1), comprising the states of Victoria (VIC), New South Wales (NSW), South Australia (SA), and Tasmania (TAS), are associated with slow-moving transient anticyclonic upper-level potential vorticity (PV) anomalies over the Tasman Sea (e.g., Marshall et al., 2013; Parker et al., 2014a; Quinting et al., 2017; Parker et al., 2019). The anticyclonic PV anomalies and the associated subsidence drive heatwaves over VIC (Parker et al., 2014b; Quinting et al., 2017). Several mechanisms can lead to the formation of anticyclonic PV anomalies. One such mechanism is the

excitation and propagation of synoptic-scale Rossby wave packets (RWP). These RWPs are often initiated several days before the onset of the heatwaves, but they amplify, and eventually break over SEA as anticyclonic equatorward (LC1-type) Rossby wave breaking (Parker et al., 2014a; O'Brien and Reeder, 2017).

Surface temperature anomalies associated with transient RWPs form, amplify, and decay within synoptic timescales, but the recurrence of RWPs in the same phase on a sub-seasonal timescale can result in persistent surface weather conditions by

repeatedly re-enforcing the surface temperature anomalies (e.g; Hoskins and Sardeshmukh, 1987; Davies, 2015). Röthlisberger et al. (2019) termed this phenomenon "Recurrent Rossby wave packets" (RRWPs) and demonstrated a

statistically significant connection between RRWPs and persistent surface temperature anomalies on a climatological timescale in the Northern Hemisphere (NH). Ali et al. (2021) found that RRWPs are also associated with persistent dry and wet spells in several regions across the globe. There is much to learn about mechanisms generating RRWPs and the interaction of RRWPs with other phenomena acting on different timescales.

Surface temperature anomalies in the extratropics are also well-known to be associated with slow-moving (stationary) blocking anticyclones. From a PV perspective, blocks are regions of anticyclonic PV anomalies in the upper troposphere large enough to disrupt the westerly jet stream and flanked by cyclonic PV anomalies (e.g., Hoskins et al., 1983; Schwierz et al., 2004; Nakamura and Hang, 2018). Blocks have been identified as a cause of major heatwaves in the NH midlatitudes (e.g., Black et al., 2004; Pfahl and Wernli, 2012; Schneidereit et al., 2012; Drouard and Woollings, 2018) because they lead to positive surface temperature anomalies due to clear-sky conditions and subsidence mainly in the central part of the anticyclone (Trigo et al., 2004; Pfahl and Wernli, 2012).

RRWPs and blocking anticyclones are closely linked. Röthlisberger et al. (2019, their Fig. 11) postulated three types of interactions between RRWPs and atmospheric blocking in the NH: blocking at the downstream end of an RRWP with the transient waves fuelling the block (Shutts, 1983); blocking acting as a metronome, setting up recurrent phasing of the waves downstream; and a combination of both mechanisms. Röthlisberger et al. (2019) showed that RRWPs often co-occur with blocking in the North Atlantic and North Pacific basins.

Finally, quasi-stationary anticyclones, linked to amplified and longitudinally elongated Rossby waves, have also been observed in association with quasi-resonant wave amplification (QRA) events (Petoukhov et al., 2013). During QRA events, synoptic-scale free waves and forced waves interact, which non-linearly amplifies the wave amplitude (see Kornhuber et al., 2017a for more details on QRA). Based on the approximations of linear Rossby wave theory, the QRA framework has so far been tested with reanalysis data and currently awaits further validation from more idealized modelling frameworks (e.g., Mooring & Linz 2020). QRA conditions have been diagnosed for several recent heatwave events: the Russian heatwave of 2010 (Petoukhov et al., 2013), and the heatwaves of summer 2018 in the NH midlatitudes (Kornhuber et al., 2019). Kornhuber et al. (2017b) presented evidence from reanalysis data for QRA of wavenumbers 4 and 5 in the Southern Hemisphere (SH). QRA is one of the mechanisms suggested to foster multiple simultaneous blocking events that are linked

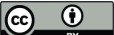

to slow-moving amplified Rossby waves of synoptic wavenumber ($k \geq 5$) (Kornhuber et al., 2017a; Petoukhov et al., 2013). However, Wirth and Polster (2021) suggest an inverse causal link by which blocking could create the waveguide structures that are required to identify QRA conditions. The links and potential causality between QRA and blocking hence, remains an

open question.

Transient high-frequency recurrent Rossby wave packets (RRWPs) have also been observed for some heatwave events linked to the QRA mechanism in the NH: The 1994 European heatwave identified as a QRA event in Kornhuber et al. (2017a) has also been identified as an RRWP event in Röthlisberger et al. (2019), and the Russian heatwave of 2010 identified as a QRA event has been shown to be composed of RRWPs (see Fig. 10 in Fragkoulidis et al., 2018). The co-

occurrence of QRA and RRWP in these cases suggests an organization of transient non-stationary wave packets during QRA conditions. The 7–15 day time-filter applied in the QRA-framework to define the background-state might obscure a substructure that becomes apparent on sub-daily Hovmöller diagrams. However, the climatological frequency of co-occurrence of QRA and RRWPs is unknown, as are potential interactions between the two. Thus, studying the dynamical processes acting on weather timescales during QRA conditions is of interest to uncover the potential interactions with

RRWPs.

The examples above highlight the need to assess how the transient features captured by the RRWP framework interact with stationary features, such as blocking and amplified Rossby waves that occur regionally or at a hemispheric scale. We investigate the link between these dynamical frameworks with a primary RRWP standpoint using co-occurrence and composite analyses. We use climatological datasets of blocking anticyclones (Schwierz et al., 2004; Rohrer et al., 2018),

QRA conditions (Kornhuber et al., 2017b; Petoukhov et al., 2013), and RRWPs (Röthlisberger et al., 2019; Ali et al., 2021) to quantify the co-occurrence of the three features and mechanisms. The data sets also allow us to investigate some of the proposed causal links discussed above and summarized in Fig. 2.




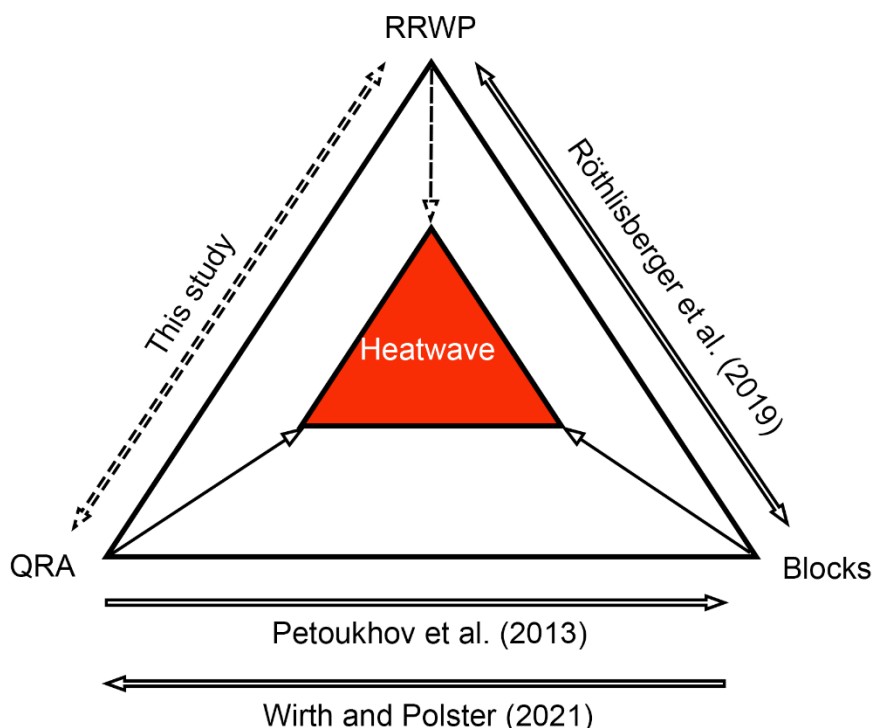

**Figure 2: Schematic summary of the links between QRA conditions, blocking, and RRWPs with heatwaves. Solid arrows indicate**
**links reported in the literature, and dashed lines indicate unexplored links.**

This paper follows the following sequence: first, we extend Röthlisberger et al.'s (2019) analysis to study the relevance of

RRWPs for persistent hot spells in the SH on a climatological time-scale with a particular focus on SEA. We then focus on

SEA heatwaves and study the co-occurrence of RRWPs, QRA, and blocks in two cases. Subsequently, we shift our attention

to RRWPs, QRA, and blocks and investigate their interactions on a climatological scale. Thus, this paper addresses four

research questions:

- Are RRWPs relevant for persistent hot spells in the SH and if so, in which regions?

- How do SH RRWPs relate to Australian heatwaves, and do QRA conditions and blocks play a role?

- How do RRWPs conditions relate to QRA conditions in the SH?

- How do RRWPs and QRA conditions relate to blocks in the SH?





## 2. Methods

### 2.1 Data

This study uses ERA-Interim (ERA-I) reanalysis data (Dee et al., 2011) provided by the European Centre for Medium-Range Weather Forecasts on a $1° \times 1°$ spatial grid for 1979–2018. Various fields are used including horizontal velocity, meridional velocity, 2 m temperature, PV, and sea surface temperature (SST). The datasets are freely available to download from https://apps.ecmwf.int/datasets/data/interim-full-daily/levtype=pl/. Note that PV fields in the SH are multiplied by a factor of -1. The climatological mean is calculated with respect to the period 1980–2010.

### 2.1 Recurrent Rossby Waves

The metric $R$ developed by Röthlisberger et al. (2019) is used identify recurrence of synoptic-scale Rossby wave patterns. For the SH, we use the same metric as in Ali et al. (2021). First, a 14.25 day running mean of meridional velocity fields $(\hat{v}_{tf}(\lambda, t))$, averaged between 35° S and 65° S, are calculated to isolate signals with timescales longer than the synoptic timescale for each longitude $\lambda$ and time $t$. The envelope of the synoptic wavenumber contribution to the time-filtered $v$ is extracted following Zimin et al., (2003). To do this, the time-filtered $v$ fields are transformed into the frequency domain using a fast Fourier transform over longitude, $\hat{v}_{tf}(k, t)$. Finally, an inverse Fourier transform is applied to calculate the envelope of the wave while only considering contributions from a selected band of synoptic wavenumbers $k = 4$–$15$. Thus, $R(\lambda, t)$ for each longitude $\lambda$ and time $t$ is calculated as

$$R(\lambda, t) = \left| \sum_{k=4}^{k=15} \hat{v}_{tf}(k, t) e^{2\pi i k l_\lambda / N} \right| \tag{1}$$

Fig. A1 shows day-of-year climatology of the $R$ metric in the Southern Hemisphere and compares it to that of the Northern Hemisphere.

High $R$ days are defined as days on which the zonal mean $R$ is greater than the 85th percentile. The code for calculating $R$ metric is freely available (check Code and data availability section).

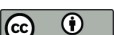

## 2.2 Atmospheric blocks

Atmospheric blocks are identified from persistent anticyclonic PV anomalies averaged between 500 hPa and 150 hPa vertical levels with the detection scheme described in Schwierz et al. (2004) as updated by Rohrer et al. (2018). The code

used is available on GitHub (https://github.com/marco-rohrer/TM2D). The detection scheme uses a 1.3 PVU threshold, a persistence criterion of 5 days, and a minimum overlap of 0.7 between two timesteps. Blocking fields identified with this algorithm are available at 6 hourly temporal resolution and $1° \times 1°$ spatial resolution. Blocking fields are resampled into daily fields for further analysis. We tested the sensitivity of the blocking fields with a 1.0 PVU threshold for the two case studies.

## 2.3 QRA data

QRA events are identified using the QRA detection scheme described in Kornhuber et al. (2017a), based on ERA–I daily fields for December to February 1979–2018 at a spatial resolution of $2.5° \times 2.5°$, smoothened using a 15 day running mean. The detection scheme tests climate data for the resonance conditions defined by Petoukhov et al. (2013): the formation of a wave guide in the zonally averaged zonal wind field for a wavenumber $k$ and the emergence of a forcing pattern of

wavenumber $m \approx k$. Please refer to Kornhuber et al. (2017a) and Kornhuber et al. (2017b) for more details. For the co-occurrence analysis and the composites, we use the period of December to February 1979–2018, for which QRA data is available. For simplicity, days with QRA conditions are referred to as QRA days and those without as non–QRA days. 819 days out of 3520 days show QRA conditions, 576 of which show QRA with wavenumber 4.

Note: Here, QRA implies that the condition of waveguide and forcing is fulfilled and a high amplitude wave is observed.

## 2.4 Southeast Australian Heatwaves

A station-based heatwave dataset is used to focus on extreme and persistent heatwaves in SEA to study the links between RRWPs, blocks, and QRA conditions. Following the methods developed in Parker et al. (2014a) and refined in Quinting et al. (2017), heatwaves in SEA in December–February (DJF) are detected from temperatures observed at the Australian Bureau of Meteorology's (BoM) monitoring stations (Fig. 1). The BoM's Australian Climate Observations Reference





Network – Surface Air Temperature (ACORN-SAT, available at _http://www.bom.gov.au/climate/data/acorn-sat/#tabs=ACORN%E2%80%90SAT_) is a high-quality temperature dataset used to monitor long-term temperature trends. The dataset provides a daily maximum temperature (TMAX) for each station. These TMAXs are extracted for stations in SEA as defined here, for DJF from 1979 to 2019. The 90th percentile TMAX (T90) is then calculated for each station for each month in DJF. A heatwave is defined as any period of at least four consecutive days for which the TMAXs at three or more

of these stations equal or exceed the T90 for that station and month. From here on, the term "heatwave" refers to heatwave in SEA. This criterion results in 58 heatwaves, which were on an average 8 days long with the longest 22 days.

### 2.5 Hot spells in the SH

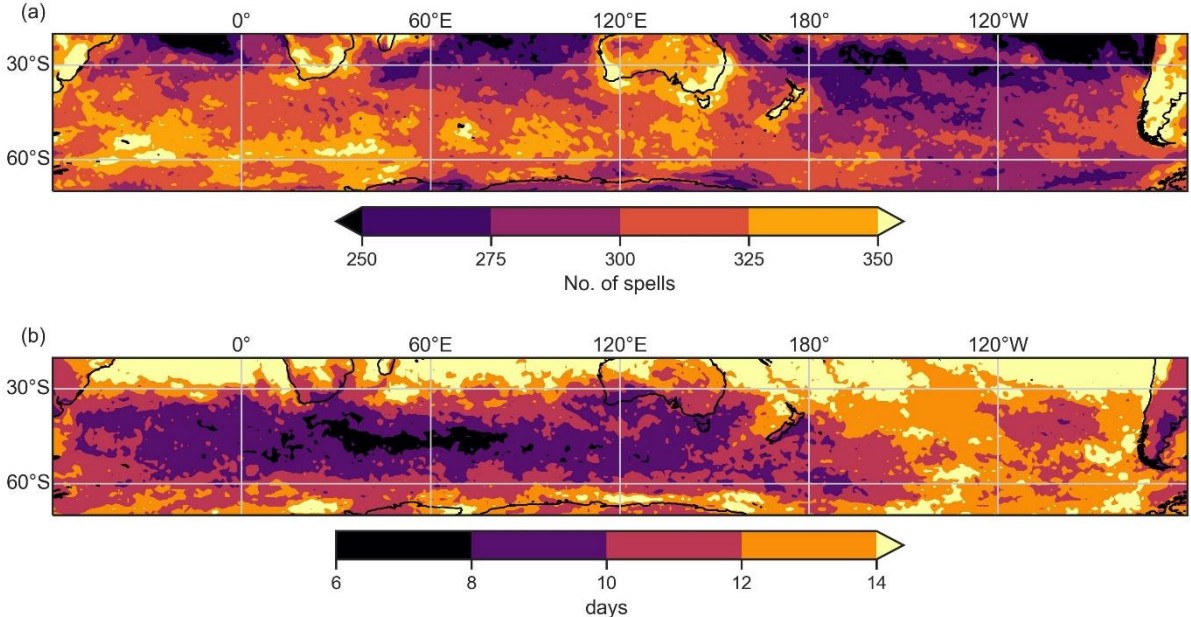

**Figure 3: (a) Total number of hot spells in November–April identified at each grid point between 20° S and 70° S. (b) The 95th**
**percentile of hot spell durations.**

2 metre temperatures (T2M) from the ERA-I fields at 6 hourly temporal resolution and 1degree spatial resolution between 20°S and 70°S are used to identify hot spell durations for 1980–2016. The hot spells definition follows that of Röthlisberger et al. (2019), in which a hot spell is calculated for each grid point as consecutive values exceeding the 85th percentile from the linearly detrended T2M fields. Spells separated by less than a day are merged to form a single uninterrupted spell. Spell

durations of less than 36 hours are excluded from further analysis. We identify hot spells for the period of November to April





as this allows longer spells, which are required for the model to work. Figure 3a shows the spatial distribution of the number of hot spells at each grid point between 20° S and 70° S. Higher number of hot spells are seen over land where parts of SEA, South Africa, and South America show 350 or more spells. The 95[th] percentile for hot spell duration varies from 6 days to more than 2 weeks (Fig. 3b). Over SEA, the 95[th] percentile duration varies from a week to roughly 2 weeks.

**2.6 Weibull regression model to assess the effect of RRWPs in the SH hot spells**

We extend the analysis of the effect of RRWPs on hot spell durations from Röthlisberger et al. (2019) to the SH, including SEA, using the same Weibull regression model. This model allows us to model the duration of the hot spells at each grid point as opposed to classifying binary information about the occurrence of heatwaves (e.g., over SEA) based on a predictor (e.g., RRWPs). Another advantage of Röthlisberger et al.'s (2019) model is that we do not need to subjectively define the

duration of a significant spell because the model accounts for the assessment of changes in all quantiles of the spell duration modelled. The null hypothesis tested here is that RRWPs have no effect on the duration of hot spells. The Weibull model is only briefly introduced here. Please refer to Röthlisberger et al. (2019) for further details and their SI for a detailed introduction to the Weibull model.

For each hot spell $i$ at grid point $g$ with a duration $D_{g,i}$, the raw $R$ metric $R(\lambda, t)$ is longitudinally averaged within a 60°

longitudinal sector centred at the grid point g with longitude $\lambda_g$ to yield $R_{lon}(\lambda, t)$. Then, a median of $R_{lon}(\lambda, t)$ is calculated for the lifetime of the hot spell to assign a representative value of $R$ ($\tilde{R}_{\lambda_g,i}$) for each spell. Thus, our model is given as (see Röthlisberger et al., 2019 for further details)

$$\ln(D_{g,i}) = \alpha_{0,g} + \alpha_{1,g}\tilde{R}_{\lambda_g,i} + \sum_{j=2}^{6} \alpha_{j,g} m_j(t_{g,i}^{start}) + \sigma_g \epsilon_{g,i} \quad ; i = 1, \dots n_g. \tag{2}$$

This model is fitted to durations of hot spells at each grid point. It results in a spatial field of regression coefficients $\alpha_{j,g}, j =$

$0, \dots, 6$, together with their $p$ values. Here, $\alpha_{1,g}$ represents the effect of $\tilde{R}$ on the hot spell duration. The $\exp(\alpha_1)$, referred to as the acceleration factor ($AF$), corresponds to the factor of change in all quantiles of the spell duration $D$ per unit increase in $\tilde{R}$ (Hosmer et al., 2008; Röthlisberger et al., 2019; Zhang, 2016). Statistical significance of $AF$ is evaluated by applying the



false discovery rate (FDR) test of Benjamini and Hochberg (1995) at maximum FDR 0.1 as in Röthlisberger et al. (2019).

Thus, regions with $AF>1$ ($AF<1$) experience an increase (decrease) in spell duration with increasing (decreasing) $R$.

## 2.7 Controlling False Discovery Rate (FDR)

We use Benjamini and Hochberg's (1995) procedure to control for type I errors due to falsely rejecting the null hypothesis in multiple independent tests. For the statistical tests used in the composite analysis, the FDR threshold, $\alpha_{FDR}$ is set to 0.1, where FDR is the expected ratio of the number of false positive discoveries to the total number of discoveries: rejection of the null hypothesis. We choose $\alpha_{FDR}$ as $\alpha_{FDR} = 2\alpha_{global}$ as per Wilks's (2016) recommendation, where $\alpha_{global} = 0.05$ is the significance level chosen for the statistical tests in the composite analysis.

## 3. Results

### 3.1 Climatological effect of RRWPs on hot spell duration in Australia

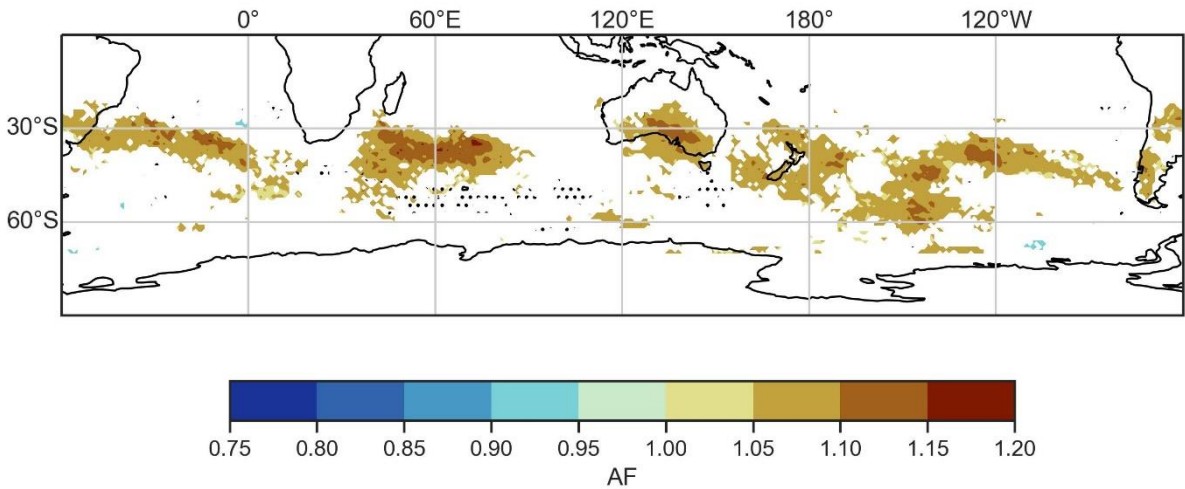

**Figure 4: Statistically significant acceleration factors (AF) for hot spells in November–April between 20° S and 70° S. Colours show AFs from a Weibull model with R metric as a covariate. Stippling indicates areas where spell duration does not follow the Weibull model based on the Anderson–Darling test at a significance level of 0.01.**

RRWPs have a significant effect on the duration of hot spells in several regions within the SH and in particular over SEA (Fig. 4). Recall that $AF > 1$ ($AF < 1$) means that an increase or decrease in $R$ is related to an increase or decrease in hot spell duration, respectively. Thus, several parts of central and southern Australia, including the states of SA, VIC, NSW, and TAS,

experience longer hot spells during periods when RRWPs occur. Interestingly, Northern Australia does not show such an association, which agrees with previous studies showing different dynamical pathways for Northern and Southern Australian heatwaves (Parker et al., 2019; Quinting et al., 2017). Other statistically significant areas over land include parts of South America: southern Brazil, Bolivia, and parts of Argentina and Chile. The $AFs$ in Fig. 4 also form a wavenumber 4 spatial pattern, in contrast to the wavenumber 7-8 pattern in the NH summer half-year (MJJASO) seen in Röthlisberger et al.

(2019). However, the presence of a spatial pattern does not necessarily indicate the existence of a dominant circumhemispheric wave during RRWPs. It merely highlights areas where the transient waves building up the RRWPs have a predominant phasing. The regression analysis shows that RRWPs are an important feature in the SH as well: RRWPs increase the duration of hot spells in several regions over land, including SEA; however, the analysis does not provide any information about how frequently RRWPs and SEA heatwaves coincide. Accordingly, we next focus on SEA heatwaves

using an observation-based dataset and quantify the simultaneous presence of RRWPs, QRA conditions, and atmospheric blocking.

### 3.2 RRWPs, Blocks. and QRA during two extreme and persistent SEA heatwaves

#### 3.2.1 Case 1: 2004 Heatwave

The February 2004 heatwave (7–22 February) lasted for 16 days. More than 60% of continental Australia recorded

temperatures above 39°C during this event (National Climate Centre, 2004). At the time, this event was the most severe February heatwave on record in both spatial and temporal extent and ranked in the top five Australian heatwaves for any month (National Climate Centre, 2004). More than 100 stations in SA, NSW, and northern VIC experienced record temperatures for February, and in some regions all-time records were set for consecutive days of heat (BoM, 2004).





**Figure 5: RRWPs, blocks, and QRA conditions during 2004 SEA heatwave. (a) Filled contours depict mean of standardized anomalies of daily maximum 2 m temperature over land for the duration of the heatwave. Contours show mean blocking frequency during the heat wave (5, 10, 20%). (b) Bars show daily maximum 2 m temperature averaged over SEA (°C); red marks the heatwave period. The Hovmöller diagram shows the meridional wind at 250 hPa averaged between 35°S and 65°S (filled contours, m/s), *R* values (grey contours, 6, 8, 10 m/s), and longitudes at which at least one grid point between 40° S and 70° S featured an atmospheric block (stippling). Rossby wave trains (blocks) are labelled in Magenta (black). The right columns in (b) indicate the presence of waveguides and quasi-resonance amplification conditions, with coloured dots indicating the wavenumber (see legend).**



**Figure 6: (a), (b), (c), (d), (e) show meridional velocity at 250 hPa (colour shading), 2 PVU contours at isentropes 340 K (black line) and 350 K (grey line) at various time steps. Stippling and orange contours show blocks identified using a 1.3 and 1.0 PVU threshold, respectively. (f) shows standardized SST anomalies with respect to austral summer (DJF) climatology for heatwave days, including the 10 days prior to the heatwaves (27 January–22 February).**



The flow was highly non-linear upstream of South America in the Pacific basin before the onset of the heatwave. On 26 January 2004, prior to the onset, a split jet structure was associated with a block south of Australia, where the subtropical branch of the jet was located over SEA (Fig. 6a). On 30 January, an anticyclonic wave breaking (AWB) event occurred upstream of Australia in the Indian Ocean at ~100° E. A further AWB event took place on 3– 5 February over SEA. The PV fluxes associated with the wave breaking helped to shift the jet southward over the Tasman Sea and in the formation of the first ridge (Fig. 6b). The flow over Australia became zonal again on 7 February and remained so until 9 February, when a short-wave ridge passed over SEA (not shown). On 10 February, AWB over western Australia resulted in the amplification of a subsynoptic to synoptic-scale ridge downstream over SEA, and a large-scale AWB over the southern Indian Ocean (~30° E) led to the formation of a downstream ridge across most of the southeastern Indian Ocean. Simultaneously, the sea surface temperatures (SSTs) were anomalously warm in the Indian Ocean. In fact, SSTs were anomalously high in several parts of the SH, including parts of the Pacific and the Atlantic Ocean for the whole duration of the heat wave as well as 10 days prior to the onset of the heatwave (Fig. 6f).

Next, a synoptic-scale wave arrived over the southern Indian Ocean, and a ridge began to form over Australia on 12 February as part of a transient and nonstationary RWT (T1 in Fig. 5b). On 13 February, conditions for quasi-resonance were met for wavenumber 5 (Fig. 5b), which may have led to the amplified waves around most of the Southern Hemisphere (Fig. 6c, 6d). Two further ridges formed over SEA on 16 and 18 February (Fig. 6c, 6d), each ridge being part of a transient nonstationary RWT. This series of upper-level recurrent ridges was part of the RRWPs and contributed to the persistence of the heatwave. These recurrent ridges associated with RRWPs were also detected by the $R$ metric (grey contours in Fig. 5b). By 20 February, the flow had returned to zonal over SEA, but a split jet formed over SEA by 22 February.

No blocks were identified directly over SEA during the heatwave, but blocks were present south of SEA and further downstream (Fig. 5, 6). The RWT labelled as T1 in Fig. 5b formed downstream of a block (B1 in Fig. 5b) south of South America around 7 February. Another RWT (T2 in Fig. 5b) seems to have been set off by a block over the Indian Ocean (B4 in Fig. 5b). Simultaneously, another block was present south of South America (B2 in Fig. 5b, 6d), and they seem to set off another RWT (T3 in Fig. 5b). Block B4 also resulted in the amplification of the Rossby wave over the Indian Ocean on 16 February (Fig. 6d). Thus, we argue that blocks could have played a key role in the initiating, phasing, and meridional

amplification of the three Rossby wave trains (T1–T3) that reached Australia between 13 and 18 February. In summary, we

saw three RWTs that passed over Australia during the QRA period (Fig. 5b). These waves were not stationary, they were not

triggered in the same area, and they were not in phase upstream of Australia.

### 3.2.2 Case 2: 2009 Heatwave

The 2009 heatwave (27 January–9 February), although extensively covered in literature (e.g., Engel et al. 2013, Parker et al.

2014b), has been chosen because it is one of the most severe heatwaves in SEA. It lasted for 14 days. During 28–31 January

and 6–8 February, temperatures in SEA were exceptionally high. On Black Saturday, 7 February, the hot, dry, and windy

conditions worsened many catastrophic fires in VIC, which recorded 173 fatalities, and more than 2133 houses were

destroyed (Karoly 2009; Parker et al., 2014b; VBRC 2010).

Prior to the onset of the heatwave, the large-scale upper-level flow was highly non-linear, with several basin-wide AWB

events in the south Pacific and south Atlantic similar to the 2004 heatwave. However, unlike the 2004 heatwave, the flow

was not zonal over SEA prior to the heatwave, with an AWB occurring on 17 January over SEA in association with a block

present upstream over the South Atlantic. A hemisphere-wide amplified Rossby wave was present during 17–19 January

(Fig. 7b); however, the forcing conditions for QRA were not met.

RRWPs occurred from 26 January onwards with an upper-level ridge forming over Australia (Fig. 7b, 8b). An AWB

occurred over SEA, resulting in an anticyclonic PV anomaly: a ridge at upper levels over SEA (see Parker et al., 2014 for a

detailed analysis of this event). The downstream edge of the ridge was located over SEA with the potential for quasi-

geostrophic forced subsidence. On 2 February, a new ridge formed over eastern Australia (Fig. 8c). On 5 February, yet

another ridge formed over SEA (Fig. 8d) and remained stationary until the end of the heatwave on 9 February.

No blocks were identified directly over SEA during the heatwave (Fig. 7, 8). Blocks were present throughout the heatwave

upstream of SEA from 50° E to 70° E in the Indian Ocean (B2 in Fig. 7b), and downstream of SEA from 200° E to 250° E

(B1 in Fig. 7b, 8). Three of the RWTs passing over Australia (T1, T2, T3 in Fig. 7b) seem to have formed downstream of the

blocks in the Indian Ocean (Fig. 7b). Multiple Rossby wave breaking events were also associated with the RWTs during this





Figure 7: Same as in Fig. 5 but for February 2009 SEA heatwave.





**Figure 8: Same as in Fig. 6 except for 2009 SEA heatwave.**





period. In the next section, we assess the importance of RRWPs and QRA conditions for all the SEA heatwaves in the period

1979–2018 and establish whether they increase the probability of heatwaves or not.

## 3.3 Are heatwaves more likely to occur with RRWPs or QRA?

| | | High $R_{SEA}$ | QRA-all | QRA ($k = 4$) | QRA ($k = 5$) |
|---|---|---|---|---|---|
| SEA heatwave days | 461 | 93 | 128 | 85 | 51 |
| SEA nonheatwave days | 3059 | 435 | 691 | 491 | 154 |
| **Total** | 3520 | 528 | 819 | 576 | 205 |
| P (heatwave \| $R$ or QRA) % | $P_{heatwave} = 13$ | 17.6 | 15.6 | 14.8 | 24.9 |

**Table 1: Occurrence of High R and QRA on SEA heatwave days and the associated conditional probabilities of a heatwave given**
**high R or a QRA day; QRA-all considers wavenumbers 4 to 6.**

QRA with $k = 5$ has the highest conditional probability of heatwave: 25 %, implying that out of 205 days with QRA

conditions, 51 are heatwave days (51/205 = 0.25). Both High $R_{SEA}$ and QRA, including all wavenumbers, have conditional

probability of heatwave greater than the reference climatology (Table 1). Heatwaves are 1.35 times more likely during high

$R_{SEA}$ days and 2 times more likely during QRA days with k = 5. Parker et al. (2014a) assessed the conditional probability of

heatwave over the state of Victoria for various large-scale drivers. They found the highest conditional probability of

Victorian heatwaves (12%) for Madden–Julian oscillation (MJO) phase 4. However, our results are not directly comparable

to Parker et al.'s (2014a), because we define heatwaves over a larger region targeting SEA and use a slightly different

heatwave definition. Given the importance of RRWPs and QRA over SEA, we next show their co-occurrence on a

climatological timescale and diagnose their association with each other and with atmospheric blocks.

## 3.4 RRWPs, Atmospheric Blocking, and QRA

### 3.4.1 Are RRWPs and QRA events independent?

The 2004 heatwave featured co-occurrence between RRWPs and QRA. This raises the question to what degree the two

phenomena are exclusive or whether the metrics capture essentially the same flow structures but interpret them differently.

Therefore, we examine the co-occurrence of high $R$ events and QRA events on a climatological scale (December 1979–





February 2018). Because QRA is a hemispheric-scale metric, we consider a zonal mean $R$ for a representation of

hemisphere-wide recurrence. However, since RRWPs can occur locally, we performed sensitivity tests using zonal max $R$ to

define high $R$ days. The zonal max approach did not change our conclusions.

|  | QRA | No QRA | **Total** |
|---|---|---|---|
| High $R$ | 25 | 11 | 36 |
| No high $R$ | 29 | 170 | 199 |
| **Total** | 54 | 181 | 235 |
| **Odds Ratio** | $\dfrac{25 \times 170}{29 \times 11} = 13.32$ | | |

**Table 2. Contingency table for high $R$ and QRA events in DJF. Note that the table only includes independent events.**

We find that 331 of 528 high $R$ days (63%) correspond to QRA days. Conversely, 331 of 819 QRA days (40%) correspond

to high $R$ days. However, these two metrics are highly auto correlated as they use 15 day running mean fields. Thus, to test

the association between the two metrics, we use every 15[th] observation to have independent events. The co-occurrences of

the two metrics are summarized in a contingency table (Table 2). We test the association between high $R$ and QRA events

using a chi-square test at 99% threshold, where the null hypothesis is: there is no association between high $R$ and QRA

events. The odds ratio is used to quantify the strength of association between QRA and high $R$ events. The odds of a QRA

event given high $R$ is 25/11. The odds of a QRA event given no high $R$ is 29/170.

The odds ratio is thus $\frac{25/11}{29/170} = 13.32$.

Thus, QRA has higher odds of occurring with high $R$ events than without high $R$ events. The chi-square test also shows a

significant association between the two, suggesting that the null hypothesis that there is no association between high $R$ and

QRA events can be rejected. We find that our test results are robust with respect to the starting step of the 15 day intervals.

In figure 9, we compare the zonal spread of mean $R$ values for different samples: high $R$ (528 days), QRA (819 days), high $R$

but not QRA (197 days), and QRA but not high $R$ (488 days). Highest recurrence is seen over the south Pacific Ocean for all

the samples. The sample belonging to high $R$ but not QRA days shows the highest mean values probably due to the smallest

sample size. QRA days show higher mean $R$ than DJF mean as 40 % of QRA days are comprised of high $R$ days. Removing

the high $R$ days from the QRA sample drops the mean $R$ values making it indistinguishable from the DJF mean.



The frequency analysis indicates a reasonably strong association between high *R* and QRA conditions but does not alone

offer any meteorological interpretation. Therefore, we calculate composites of tropopause-level potential vorticity (PV) and

zonal winds for high *R* and QRA days (Fig. 10).

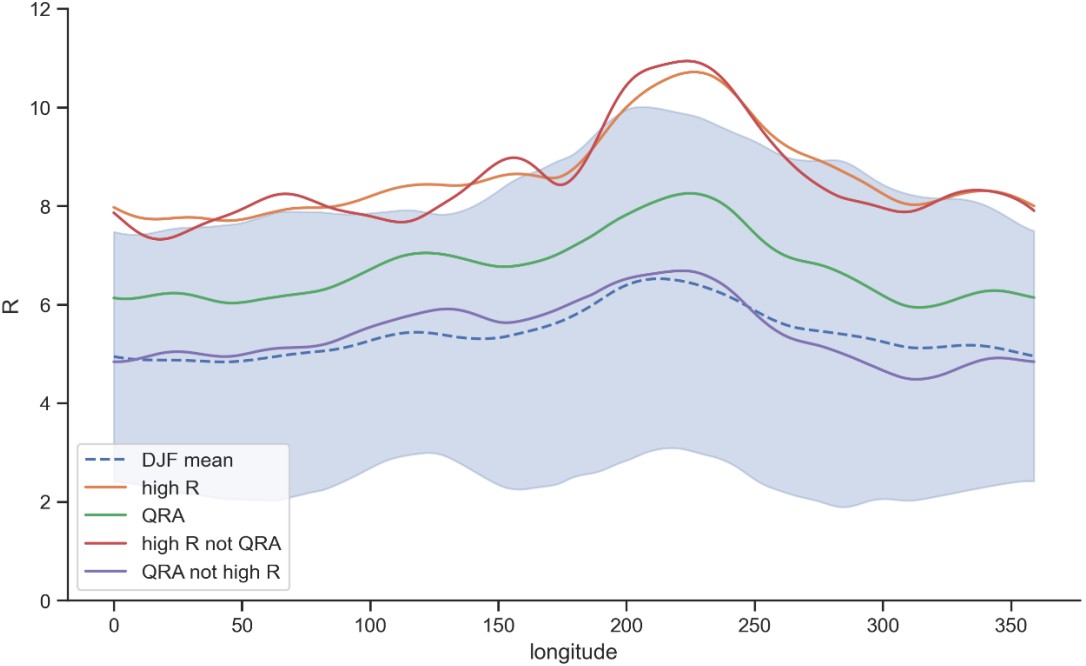

**Figure 9: A comparison of mean *R* values for different samples shown in legend. Shading shows *R* values within one standard**
**deviation for DJF.**

### 3.4.2 How similar or different are upper-level flow conditions during RRWPs vs QRA days?

The anomaly structures for PV at 350 K and zonal wind at 250 hPa (U250) show a remarkable similarity between high *R* and

QRA condition days: a Pearson correlation coefficient of almost 1 between high *R* days and QRA days for composite mean

PV and composite mean U250 fields, respectively. Anomalies of PV composites with respect to the DJF climatology for

both QRA and high *R* days (Fig. 10a, 10b) feature cyclonic PV anomalies north of New Zealand, in the Pacific Ocean, and

upstream of South America. Similarly, anticyclonic PV anomalies are present in all the major ocean basins, including

downstream of Australia, upstream of South America, and both upstream and downstream of South Africa. Increase in

blocking frequency (crossed hatches in Fig 10a, 10b) is also seen mainly over south Pacific Ocean. Similarly, spatial features

of the U250 anomalies (Fig. 10d, 10e) show a striking similarity between the two composites, with stronger westerlies over



the South Atlantic Ocean and weaker zonal winds over SEA. However, a key difference is visible upstream of South America, where high $R$ days show positive U250 anomalies north and south of the climatological jet core and negative U250 anomalies at the climatological jet core, thus favouring a meandering jet. The major features in the spatial distribution of anomalies for high $R$ days are robust when testing the sensitivity by defining high $R$ days with zonal maximum $R$ values instead of the zonal mean $R$ values used here. This suggests that the basin-wide high $R$ values projects out in the zonal mean

fields as well.

Subsequently, we compare the sample of QRA days exclusive of high $R$ days with those of high $R$ exclusive of QRA days for PV and U250 fields respectively. The null hypothesis tested is that the two samples belong to the same distribution. A two-sample Kolmogorov–Smirnov (K–S) test at 5% threshold is used to evaluate the null hypothesis with maximum FDR at 10%. The resulting significant area, where null hypothesis is rejected, is shown with dotted hatches (Fig. 10c, 10f).

Significant areas difference in composite mean PV fields between the two samples include cyclonic PV anomalies over parts of Brazil, central Australia, and Pacific Ocean. The difference in composite mean U250 fields suggests strengthening of the westerlies and narrowing of the climatological jet (dotted isolines in Fig. 10f) over all the three ocean basins for QRA days exclusive of high $R$ days compared to high $R$ days exclusive of QRA.



**Figure 10: Anomalies of composite mean field with respect to DJF climatology for QRA (left), high R (middle). Right column show the difference of mean fields between QRA but non-high *R* days and high *R* but non-QRA days (right). (a), (b), and (c) show PV anomalies at 350 K isentrope. (d), (e), and (f) show zonal wind (U) anomalies at 250 hPa (m/s). Dashed contours in (d), (e) show isolines of mean U at 20, 30, 40 m/s. (f) show DJF mean U isolines. The solid contour in (a), (b) shows mean 2PVU at 350 K isoline for the respective variable and in (c) shows the same for QRA but non-high *R* days. The dashed contours in (a), (b), (c) show the 2PVU at 350 K contour for DJF mean climatology. Crossed (lined) hatches in (a), (b) show areas where blocking frequency anomalies (%) with respect to DJF climatology is greater (less) than 2 %. Dotted hatches in (c) and (f) show significant regions tested using K–S test at 5% threshold.**

### 3.4.3 Does blocking area increase during high *R* or QRA conditions?

To analyse whether high *R* or QRA conditions increase the area of the blocks, we compare the blocks between 40° S and 70°

S. First, the blocking area for each unique block is extracted at the time of maximum amplitude of PV and characterized with

respect to high *R* or non-high *R* and QRA or non-QRA conditions. The resulting kernel density estimation functions are





shown in Fig. 11a. The sample size for each category in the same order is 86, 445, 101, and 421. For the 101 blocks with

QRA days, 68 and 33 samples belong to QRA with wavenumber 4 (k4) and QRA with wavenumber 5 (k5), respectively.

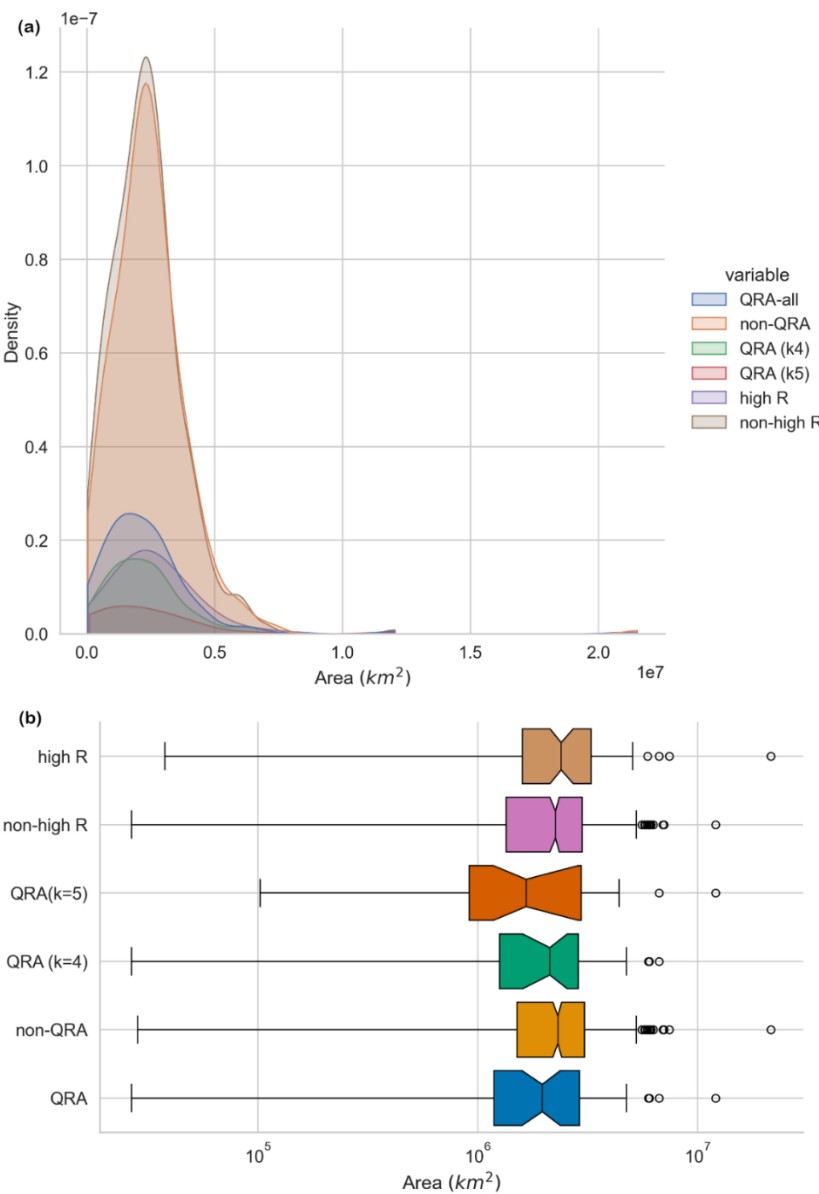

**Figure 11: A comparison of area (km²) of atmospheric blocks for various categories. (a) Kernel density estimation (KDE) of the blocking area. (b) box plot of blocking area for the categories shown in the legend.**




Density curves for the blocking area show a unimodal distribution for all the categories, with slightly right-skewed curves in most of the cases. Box plots of the blocking area under high $R$ days show a marginally higher median than non-high $R$ days whereas the blocking area under QRA days shows a marginal decrease in median compared to blocks with non-QRA days.

Whether the medians in these categories are significantly different or not is evaluated using a Mann–Whitney U test, a non-parametric test which does not require the samples to be normally distributed, using a two-sided hypothesis at 5% significance level. We did not find a significant difference in the median area of blocks for either the case of high $R$ vs non-high $R$ or QRA vs non-QRA days (Fig 11b).

### 3.4.4 How are blocks spatially distributed for high $R$ and QRA days?

Mean blocking frequency for high $R$ days (Fig. 12c) shows a statistically significant increase in blocking frequency in DJF mainly over parts of two ocean basins: upstream of South America in the Pacific Ocean, and upstream of Australia in the Indian Ocean. These are also the regions where blocking frequency is highest in the DJF climatology (Fig. 12a). Other areas with a significant increase include part of the Antarctic Sea (180°W), south of Africa, whereas a significant decrease is observed over parts of Antarctica. The significance was assessed using Mann–Whitney U test, which assesses the null

hypothesis that distribution of blocks under high $R$ (QRA) days and that of DJF climatology are equal. For QRA days (Fig. 12b), there is a significant decrease in blocking frequency over south of Australia and New Zealand, south of Africa, and over parts of the south Atlantic Ocean, which was not seen for high $R$ days. In contrast, the region upstream of South America shows an increase similar to high $R$ days. Although most of the grid points for QRA days (Fig. 12b) are not statistically significant, the overall decrease in blocking frequency is consistent with the results in Fig. 11b, which show a

slight decrease in the median blocking area for QRA days.







**Figure 12:** Blocking frequency (%) for (a) DJF, (b) difference between the mean blocking frequency (%) for QRA days and the DJF mean frequency, and (c) difference between the mean blocking frequency (%) for days with high R days present and the DJF mean frequency. Dashed lines in (b) and (c) show DJF mean blocking frequency contours drawn at 4, 6, and 8%. Dotted hatches in (b) and (c) show grid points with statistically significant difference in the composite mean to the mean of the sample in (a) assessed using a Mann–Whitney U test with FDR at 5%.





**4. Discussion**

We first summarize key criteria used in the objective identification of the weather features, blocks and RRWPs, and the

QRA mechanism. The defining spatial and temporal characteristics of blocks, QRA, and RRWPs used in their automatic

detection algorithms are summarized in Table 1. The zonal spatial scale of the structures increases from regional to basin-

wide for blocking, through regional, basin-wide to semi-hemispheric for RRWPs, to hemispheric for QRA.

| Characteristics | Blocks using Schwierz et al.'s (2004) algorithm | RRWPs using Rothlisberger et al.'s (2019) algorithm | QRA using Kornhuber et al.'s (2017b) algorithm |
|---|---|---|---|
| Input variable | PV (vertically averaged between 500-150 hPa) | Wavenumber filtered V at 250 hPa | Zonal mean U at 300 hPa for waveguide, Thermal and Orographic forcing |
| Presence of waveguide | No | No | Yes |
| Wavenumber filtering | No wavenumber filtering | $k = 4 - 15$ | Focus on $k \geq 4$ |
| Persistence/ timescale | Minimum persistence of 5 days | 14 day running mean fields | 15 day running mean fields |
| Spatial scale | Regional to basin-wide | Regional, basin-wide, or semi-hemispheric | Hemispheric |

**Table 3: A comparison of the criteria used to automatically identify blocking, QRA, and RRWPs.**

RRWPs can be regional, basin-wide, or semi-hemispheric in spatial extent. Yet, a zonal-mean approach finds that 40% of

high *R* days coincide with QRA days. This is partly because a regionally amplified pattern can influence zonal-mean fields.

For example, we found a high correlation between the high *R* days defined with zonal means with the high *R* days defined

with zonal-maximum $R$ fields. Moreover, a recurrent transient wave pattern can appear as a quasi-stationary signal when

averaged over time. Furthermore, amplified Rossby waves may influence the metrics used here for objectively detecting

blocking, RRWPs, and QRA. For example, the amplified Rossby waves during the 2004 SEA heatwave (Fig. 5), aided by

recurrence, resulted in high $R$ values, corresponded regionally to blocking, and were identified as a QRA event. These

amplified waves were clearly composed of recurring transient waves with nonzero phase velocity (Fig. 5).

The link between high $R$ and QRA events is not only reflected in high co-occurrence but also in spatial patterns. The upper-

level mean PV composites for QRA and high $R$ days show a remarkably similar pattern (Fig. 10). The pattern is statistically

indistinguishable even though the zonally averaged $R$ metric does not explicitly include any phase or location restriction.

This implies that high $R$ events in the SH co-occurring with QRA have a particular phase preference. The PV anomaly

pattern exhibits a wavenumber 4 structure in the extratropics and a wavenumber 5 to 6 pattern in the subtropics (Fig. 10).

Hence, the question is whether RRWPs and QRA may in some cases be the same structures observed through different

lenses, i.e., with diagnostics from temporal, spatial, and other filters. The similarity in the composite PV anomalies during

high $R$ and QRA days point to the same mechanisms being relevant for the organization of the RRWPs, the establishment of

QRA conditions, and/or interactions between the two.

The fact that not all high $R$ and QRA days overlap – 37% (197 days) of high $R$ days do not feature QRA conditions – may be

explained by the different longitudinal scales used by the detection algorithms. Another reason could be the forcing

condition required for QRA events; not all high amplitude waves are detected by QRA because QRA needs the presence of a

waveguide as well as thermal and orographic forcing (Petoukhov et al., 2013; Kornhuber et al., 2017b). Moreover, the

composite PV for QRA days exclusive high $R$ days showed significant differences to high $R$ days exclusive of QRA over

several key regions: parts of Brazil, central Australia, and the south Pacific Ocean (Fig. 10c).

The schematic in Fig. 12 illustrates hypotheses of interactions between RRWPs, QRA, and blocking during SEA heatwaves.

Direct interactions between RRWPs and QRA conditions might include momentum fluxes associated with Rossby wave

breaking that may establish background flow conditions conducive to QRA. External mechanisms of importance might be

planetary-scale stationary waves of the extratropical flow that organize synoptic-scale Rossby waves, and hence RRWPs,

and that may contribute to QRA conditions. These waves might be forced by sea surface temperature anomalies (e.g.,

O'Brien and Reeder, 2017) and tropical sources (e.g., Hoskins and Sardeshmukh, 1986). Tropical forcing in the form of

enhanced convection due to an active MJO was present during the 2009 heatwave (Parker et al., 2014b), for which RRWPs

were also observed (Fig. 5).

Climatological interactions between blocks and RRWPs take the form of significant increases in blocking frequency over

parts of south Pacific and Indian Ocean. Increased $R$ anomalies are seen upstream and downstream of blocks in the south

Pacific Ocean (Fig. C1), similar to the relationship shown in the Northern Hemisphere basins (Röthlisberger et al. 2019).

During SEA heatwaves, blocks were not detected directly over SEA in either of the cases that we analysed, even though

anticyclonic PV anomalies are a common feature of SEA heatwaves (Parker et al., 2014b, see also Fig. 6, 8). The

climatological blocking frequency (Fig. 12a) clearly indicates that the blocking metric primarily detects blocking further

poleward. Sensitivity tests with a slightly different setting (threshold of 1.0 PVU), which should capture blocks further

equatorward (Pfahl and Wernli, 2012), did not detect the subtropical ridges over SEA in either of the two case studies.

Moreover, the ridges over SEA were relatively transient during the 2004 and 2009 heatwaves. However, these ridges are

extremely important to identify; thus, developing algorithms to identify subtropical ridges over SEA would be beneficial

(Sousa et al., 2021).

However, blocks frequently featured upstream of Australia during the two heatwaves and played a role in initiating Rossby

waves and organizing their phases, resulting in RRWPs. Blocks were also significantly frequent in parts of Pacific and

Indian Oceans for days featuring RRWPs. The recurrent upper-level ridges associated with the RRWPs played a role in

reinforcing surface weather. QRA with $k = 5$ was observed simultaneously with RRWPs during the 2004 heatwave. Both the

cases also showed high nonlinearity in the flow as breaking Rossby waves. The PV anomalies from the resulting wave

breaking played a vital role in amplifying upstream blocks (e.g., Shuts 1983, Pelly and Hoskins 2003) and triggering Rossby

waves (e.g., Martius et al., 2010, Röthlisberger et al., 2019).



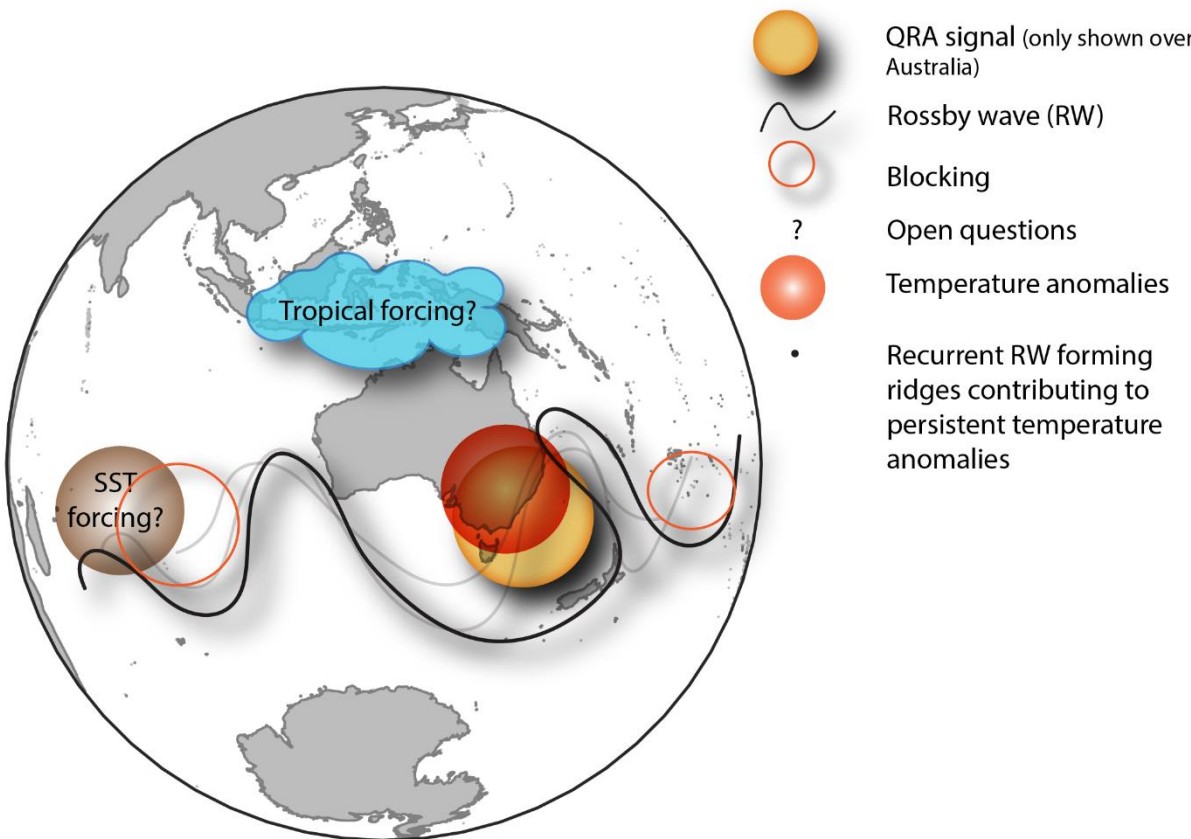


**Figure 13: Postulated schematic of interaction between atmospheric blocks, RRWPs, and QRA.**

## 5. Conclusions

To conclude, we answer the research questions put forward in this study.

Are RRWPs relevant for persistent hot spells in the SH and if so, in which regions? We saw that RRWPs increase the

duration of persistent hot spells significantly over several regions of the SH. Several parts of SEA, including the states of

South Australia, New South Wales, Victoria, and Tasmania experience longer hot spells during RRWPs. Other regions over

land include South America: southern Brazil, Bolivia, and parts of Argentina and Chile.

How do SH RRWPs relate to SEA heatwaves, and do QRA conditions and blocks play a role? We showed that both RRWPs

and QRA increase the probability of SEA heatwaves (Table 1). Heatwaves are two times more likely during QRA with

wavenumber 5 and 1.35 times more likely high $R_{SEA}$ days than reference climatology. The two case studies of the SEA

heatwaves of 2004 and 2009 showed the role of RRWPs in building persistent ridges over SEA (Fig. 5 and Fig. 7). Both

heatwaves featured RRWPs comprised of transient Rossby waves which were not in phase upstream of Australia. QRA with

wavenumber 5 was observed during the 2004 heatwave, possibly contributing to the highly amplified flow conditions (see

Fig. B1 for coincidence during 2014 heatwaves). Upper-level PV based blocks were not directly observed over SEA but

blocks upstream and downstream played an important role in initiating the Rossby wave trains. Both QRA and high $R$ days

could serve as important indicators for SEA heatwaves because they show conditional probabilities of SEA heatwaves that

are comparable with remote drivers such as MJO and El Nino (Parker et al. 2014a). However, the multiple pathways for

SEA heatwaves implies that no single metric or diagnostic tool can be perfect.

How do RRWPs conditions relate to QRA conditions in the SH? We found a strong and statistically significant association

between RRWPs and QRA events (Table 2); QRA events are more likely to occur with RRWPs than without RRWPs. We

also showed that 40% of QRA days also feature high $R$ days, indicating RRWP conditions, which implies that QRA

conditions can often feature RRWPs. We also found similar flow conditions in the composite mean upper-level fields during

QRA and high $R$ days. However, 60% of QRA days do not feature high $R$. Significant differences in flow conditions for

QRA days exclusive of high $R$ days with high $R$ days exclusive of QRA days show cyclonic PV over parts of Brazil, central

Australia, and south Pacific Ocean, and increased westerlies over all the three ocean basins.

How do RRWPs and QRA conditions relate to blocks in the SH? We found an insignificant increase in the median area of

blocks for high $R$ vs non-high $R$ in the SH and a similar decrease for QRA days vs non-QRA days (Fig. 11). We looked

further into how blocks are spatially distributed for high $R$ and QRA days, respectively (Fig. 12). Important differences can

be seen between the two in blocking frequency over the Indian Ocean, south of Australia and New Zealand, south of Africa,

and in the south Atlantic Ocean, whereas both show an increase in blocking frequency over the south Pacific Ocean.

Furthermore, we proposed a schematic of how RRWPs, QRA, and blocking could co-exist or interact (Fig. 13); however, the

hypotheses need to be tested further. Amplified Rossby waves can influence the metrics used to diagnose these features, and

thus, we detect high co-occurrences between high $R$ and QRA conditions. However, interactions between RRWPs, QRA,

and blocks may occur via momentum fluxes associated with amplified waves, amongst other pathways. There may be

potential common drivers such as enhanced MJO and SST forcing. Rossby wave breaking was frequently observed during

the two case studies, which may have initiated Rossby wave trains. These Rossby waves recurred in the same phase over Australia but were not in phase upstream. Possible unexplored factors which can modulate the phase of these waves include stationary planetary waves and blocks.

However, several questions that relate to the interaction between blocking, RRWPs and QRA remain open. Does blocking modulate the phase of Rossby waves and thus help in establishing RRWPs, or is the causal link instantaneous? What is the role of blocks during the QRA conditions, and why do we see a difference in blocking frequency between RRWPs and QRA conditions (Fig. 12)? The role of QRA in the recurrence of Rossby waves also needs to be investigated further. Investigating the role of background flow is not straightforward because defining it is a formidable problem (Wirth et al., 2018; see discussions in White et al., 2021). The interaction of RRWPs with other well-known climate oscillation patterns also needs to be investigated further. The improved understanding between the interplay of these features will help to reduce model biases and improve our confidence in future climate projections.

## Appendix A: Comparison of $R$ anomalies for Southern Hemisphere and Northern Hemisphere

Both the Southern and Northern Hemisphere $R$ fields show seasonality. Anomalies are highest for Northern Hemisphere boreal autumn and winter days. Interestingly, the Southern Hemisphere shows higher $R$ anomalies during austral summer days than winter days.

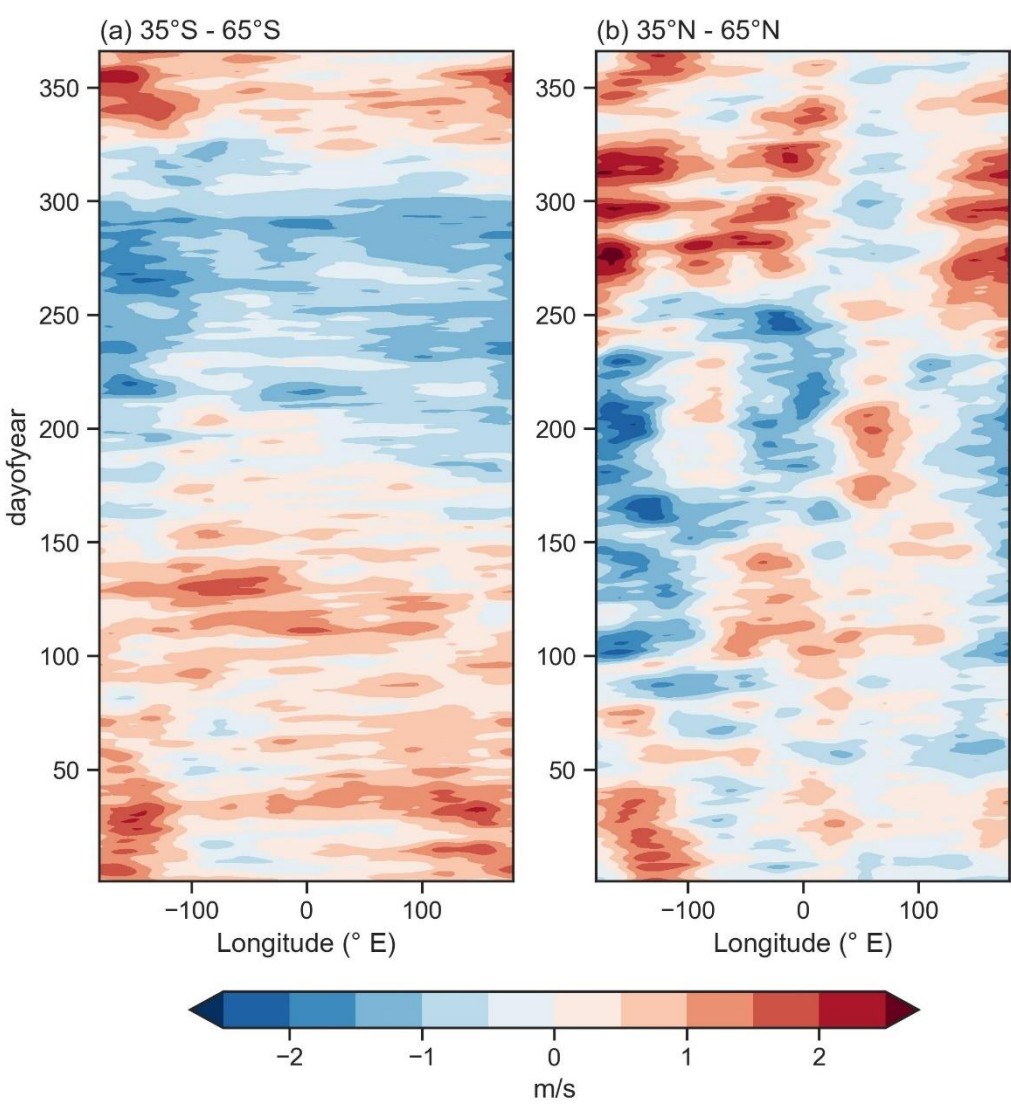

**Figure A1:** *R* **anomalies for Southern and Northern hemispheres. Anomalies for day-of-year mean are calculated with respect to mean** *R* **fields.**




**Appendix B: RRWPs and QRA during 2014 Heatwaves**

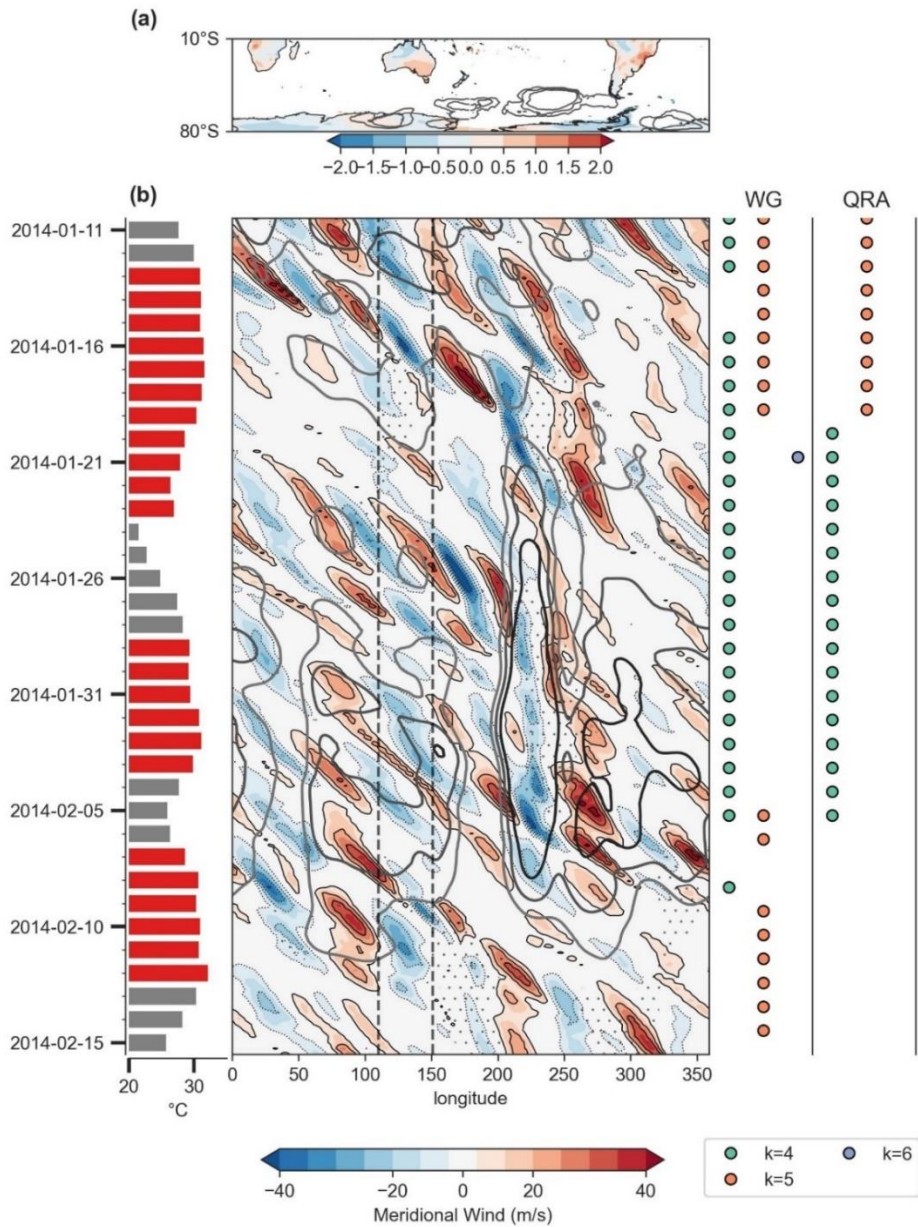


**Figure B1: RRWPs, blocks, and QRA conditions during the 2014 SEA heatwave. (a) Filled contours depict mean of standardized anomalies of daily maximum 2 m temperature over land from 2014-01-11 to 2014-02-14. Contours show mean blocking frequency during the heat wave (5, 10, 20%). (b) Bars show daily maximum 2 m temperature averaged over SEA (°C), and red marks the heatwave periods. The Hovmöller diagram shows the meridional wind at 250 hPa averaged between 35°S and 65°S (filled**
**contours, m/s), *R* values (grey contours, 6, 8, 10 m/s), and longitudes at which at least one grid point between 30°S and 70°S featured an atmospheric block (stippling). The right columns in (b) indicate the presence of waveguides and quasi-resonance conditions with coloured dots indicating the respective wavenumber (see legend).**





**Appendix C: Relationship between blocks and RRWPs in the South Pacific and the Indian Ocean**

**Figure C1: Time-lagged Hovmöller composites of R anomalies centred on the mean longitude and time of maximum amplitude of blocks located in Pacific Ocean (181–300° E, 30–80° S) in subplot (a) and (b), Indian Ocean (60–180° E, 30–80° S) in subplot (c) and (d). Left column includes blocks for all seasons and right shows for DJF. N denotes number of blocks for each category.**

To further analyse the spatial distribution of RRWPs relative to blocks in the SH, we focus on two longitudinal subdomains

that show a high blocking frequency in the DJF climatological mean (Fig. 11a): the South Pacific Ocean (230 – 310 °E), and

the Indian Ocean (0 –90 ° E). We use time-lagged composite *R* anomalies with respect to the centroid of the blocks at the

time of the maximum blocking amplitude in the two domains similar to Röthlisberger et al. (2019; see Fig. 12 in their paper). Here, $R$ anomalies are calculated with respect to the day-of-year climatology.

In the Pacific Ocean, blocks coincide with positive $R$ anomalies in a longitudinal band from ~60° upstream to ~60°
downstream of the blocks (Fig. C1 a, b) from 5 to 8 days before the time of maximum blocking amplitude; this resembles a butterfly pattern, similar to blocks in the NH (Fig. 12, Röthlisberger et al., 2019). Similar to the NH, $R$ anomalies in the Pacific Ocean are not high at the centroid of the block. This could be because the wavelength of the upper-level ridge associated with the block may be too wide to be captured by the $R$ metric because the $R$ metric only has contributions from k = 4 and higher. $R$ anomalies are consistent for DJF and blocks for all seasons in the Pacific. In contrast, in the Indian Ocean,
seasonal variation is seen in $R$ anomalies (Fig. C1 c, d), where blocks located in DJF show $R$ anomalies downstream of the centroid of the block only and possibly show weak association with RRWPs.

## Code and data availability

Code for calculating $R$ metric is available on GitHub (Ali and Röthlisberger, 2021). ACORN-SAT data is available at *http://www.bom.gov.au/climate/data/acorn-sat/#tabs=ACORN%E2%80%90SAT*. QRA data can be requested from KK. The
ERA-I reanalysis dataset used can be downloaded from https://apps.ecmwf.int/datasets/data/interim-full-daily/levtype=pl/.

## Author Contributions

OM, KK, MR, SMA conceptualized the study, and SMA and OM designed the methodology. SMA performed the formal analysis. TP provided heatwave data and contributed to heatwave analysis. KK provided QRA data.  MR provided code for the Weibull regression model and guided SMA in applying it. SMA wrote the original draft, and all the authors contributed
to its review and discussion.

## Competing Interests

The authors declare that they have no conflict of interest.



**Acknowledgements**

SMA is grateful for discussions with Alexandre Tuel and for the text editing by Simon Milligan. OM and SMA acknowledge
Marco Rohrer for the blocking algorithm. The authors acknowledge the European Centre for Medium-Range Forecasts
(ECMWF) for producing the ERA-I dataset and the Australian Bureau of Meteorology for producing ACORN-SAT dataset.

**Funding information**

SMA and OM were funded from the Swiss National Science Foundation grant number 178751. MR was funded from the
INTEXseas project from the European Research Council under the European Union's Horizon 2020 research and innovation
program (Grant Agreement 787652). KK was partially supported by the NSF project NSF AGS-1934358.



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
