# Peer review of "Recurrent Rossby waves during Southeast Australian heatwaves and links to quasi-resonant amplification and atmospheric blocks"

_Weather and Climate Dynamics, 2022_

## Editor Comment (EC1)

The initial submission of this manuscript received critical general comments from both reviewers, before the reviewers would provide more detailed reviews.

Both reviewers share fundamental reservation with the way that the authors attempt to diagnose quasi-resonant amplification (QRA). Both reviewers question the use of temperature (T) anomalies as external forcing in the QRA analysis. Reviewer#1 does so with a very basic explanation of why using T anomalies leads to a circular argument. In my own words: The very strong null hypothesis based on our fundamental understanding of midlatitude dynamics is that T anomalies within Rossby waves form *as part of* the Rossby wave by meridional advection and vertical motion (in case of a baroclinic Rossby wave). In any case: T anomalies themselves do not(!) provide a Rossby wave source. In fact, T anomalies may constitute the Rossby wave itself, e.g., as Eady edge waves or following Bretherton (1966). It is diabatic heating that provides a Rossby wave source. A strong argument is thus needed why T anomalies here could be used as a proxy for diabatic heating. In particular with the lack of land-sea contrast in the southern hemisphere (SH), it is hard for me to see how such an argument could be constructed. This aspect is crucial because with the near absence of orographic forcing in the SH midlatitudes, it is hard to see where the forcing for QRA should originate from. The response of the authors (provided to reviewer#2, who has phrased a milder version of the issue) does not sufficiently address this issue.

Reviewer#1 further questions the validity of the authors' analysis of circumglobal waveguidability, a necessary condition for QRA. The reviewer's own previous work, referred to by the authors, has shown that the method applied by the authors yields spurious results, i.e., application of the method may not only be inconclusive but may actually lead to incorrect conclusions. The authors respond that their focus is on relating south-east Australian heatwaves "to existing diagnostics presented in the literature before" and that they "use the QRA metric for only an empirical study." The authors "will clarify that the paper does not aim at further exploring the QRA mechanism itself." It is hard for me to see, however, what the empirical use of the "QRA metric" would be if not to explore the relevance of the QRA mechanism.

Given the severity of the criticism shared by two reviewers, I do not encourage to resubmit a manuscript in that the QRA analysis has been revised *according to the line of argument given in the authors' responses*. I rather suggest considering omitting this contented part from a revised version of the manuscript.

At this point, I would like to add a more general comment: Newly introduced diagnostic methods - just as other scientific results - need to stand "the test of time", i.e., the scrutiny by expert colleagues before becoming generally accepted „standard" diagnostics. The "QRA metric" is currently undergoing such scrutiny. The expectation for any diagnostic is that its application yields insight that cannot be not gained by merely looking at the data at hand. The issue raised by Wirth and Polster (2021) is not that the „QRA metric" would be a weak metric in the sense that its signal-to-noise ratio were poor. The issue raised is that the „QRA metric" yields incorrect results. If a diagnostic faces such fundamental criticism, valid arguments that nullify the criticism seem to be in order before further use of the diagnostic could be encouraged – even if the diagnostic had been used in a number of previous studies.

A further issue raised by reviewer#1 is the well-known issue of correlation vs. causation. While the authors are certainly aware of this issue, I sense in their responses that they may dismiss the issue to easily. Reviewer#1 suggests a plausible causal pathway by which a long-lasting heat wave may promote the occurrence of recurrent Rossby wave packets. The authors are clear that they "ultimately only diagnose "co-variability" ", but then they go on to argue that based on physical understanding their results do suggest a causal relation. While I much appreciate that the authors relate their statistical analysis to physical hypotheses, they do not seem to appreciate

that reviewer#1 has proposed an equally valid physical hypothesis that the authors' statistical results would support also. In statistical terms, the authors would need to control for this alternative casual pathway; and they should be aware that in the presence of confounders and colliders a simplistic correlation analysis may yield misleading results. Testing statistical significance (last paragraph of authors' response to reviewer#1) does not help with the causation vs. correlation issue.

I'd like to point out that so-called causal inference provides a rigorous mathematical framework that allows to establish cause and effect based on correlations. While this framework is not yet commonly applied in atmospheric sciences, an increasing number of studies do so, and a recent review of causal inference in Earth system science can be found in Runge et al. (2019). I do not mean to imply the recommendation that the authors use causal inference in their study – although this would certainly be interesting. I do recommend, however, that the authors do not take lightly the critique by reviewer#1 and thus are clear about their underlying physical hypotheses, discuss alternative hypotheses, and carefully word their causal interpretation of their statistical results in a revised version.

References:

Bretherton, F. P. (1966). Critical layer instability in baroclinic flows. Quarterly Journal of the Royal Meteorological Society, 92(393), 325-334.
Runge, J., Bathiany, S., Bollt, E. *et al.* Inferring causation from time series in Earth system sciences. *Nat Commun* **10,** 2553 (2019). https://doi.org/10.1038/s41467-019-10105-3

---

## Author Comment (AC1)

**Author replies**

**Reply to reviewer 1**

We thank Volkmar Wirth for taking time to evaluate this paper and appreciate the constructive remarks. We have responded to the points inline in blue font. The line numbers in the text refers to the first submitted version:

Reviewer 1

*The current paper investigates the relation between Southern Hemispheric atmospheric blocking, recurrent Rossby wave packets, so-called quasi-resonant amplification conditions, and heatwaves over Australia. The work is based on (1) investigating two relevant episodes and (2) performing a more systematic statistical evaluation using reanalysis data.*

***I have two major issues with this work.***

*( 1) First, I have a major issue with the way you try to diagnose resonance conditions. Essentially you use an algorithm from previous work, more specifically: from Kornhuber et al. 2017. This algorithm has recently been shown by Wirth and Polster (2021) to lead to spurious results: in the presence of large-amplitude waves, the algorithm is prone to diagnose two turning latitudes, which you then interpret as proof of a waveguide; however, as shown by Wirth and Polster (2021), the occurrence of two turning latitudes in the presence of large-amplitude waves is likely to be an artifact rather than an indication for a waveguide. Those who read the paper by Wirth and Polster (2021) realize that the algorithm of Kornhuber et al. (2017) is NOT appropriate to determine resonance conditions (unless you can prove otherwise). For this reason, you cannot simply quote the Wirth and Polster paper and then, for the rest of your paper, ignore their important result and continue without any further comment and detailed analysis.*

*One possibility to test whether or not the issue of Wirth and Polster applies to your case would be to repeat the analysis based on the zonalized background state instead of the zonal mean background state. If you obtain two turning latitudes with the zonalized background state, you could be somewhat reassured that this part of your analysis is free from spurious effects and, then, continue your argument. Christopher Polster from my working group would be happy to assist you with computing the zonalized background state if needed.*

*Indeed, I did refer to the Kornhuber paper "for more details" and found a second issue with the algorithm that, to my knowledge, has not been pointed out in the literature yet. In your algorithm you require a second criterion based on the amplitude of the forcing: "…the combined amplitude of the thermal and orographic forcing …. [must be] of sufficient magnitude". To be sure, orographic forcing is given and fixed and can be assumed to be "external". However, in addition to orographic forcing you use the observed temperature perturbation (= deviation from zonal mean) as a proxy*

*for a "thermal forcing" (whatever this may be in the framework of a barotropic model). But in contrast with orography, the temperature perturbation is highly "internal" and must be considered as a result of the large wave amplitudes rather than a forcing. In other words, you cannot simply compute the observed (large) temperature anomalies during episodes with large wave amplitudes and use them as "forcing" in an argument that is meant to explain the large wave amplitudes. This logic is highly circular and, therefore, meaningless.*

We want to highlight that the focus of this paper is the south-east Australian heatwaves and their relation to existing diagnostics presented in the literature before, one of which being QRA. We use the QRA metric for only an empirical study. We will clarify that the paper does not aim at further exploring the QRA mechanism itself. However, we are aware of Volkmar's recent important work on this topic and will substantially extend the discussion section to highlight it. We agree with Volkmar that further research is necessary, ideally using idealized model experiments, to explore the limitations of the current formulation of QRA conditions and develop them further. However, this is not in the scope of this paper. Therefore, we acknowledge the offer to calculate zonalised background state, but we would leave it for a more suitable outlet. We will remove Figure 13 where we suggested interactions between the three features. We will also remove other instances where we might have suggested QRA in terms of a causal mechanism and will modify the title accordingly.

On the point of thermal forcing criteria used in the QRA metric, the second reviewer had a similar comment. Please see our reply to the second reviewer's comment.

*Incidentally, on line 70 you provide a very misleading description of resonance. The phenomenon of resonance (as used in Petoukhov et al. 2013) is entirely based on linear theory, so expressions like "nonlinear amplification" and "interaction between wave A and wave B" are not in place here. In linear wave theory, you can always superimpose two solutions in order to get a new solution; generally, this leads to constructive or destructive interference, but it will never give you anything resembling resonance (see my further comments in Wirth 2020b). It would be very desirable not to perpetuate such misconceptions; rather the authors should provide a lucid description that is compatible with the fundamental concept of resonance from theoretical physics and with the early work of Haurwitz (1940) and Charney and Eliassen (1949).*

We agree that the description was not clear. Petoukhov et al. (2013) implies interference between the forced wave and free waves of similar wavelength. It is true that it is based entirely on linear theory. We will clarify the description accordingly.

*( 2) My second issue refers to the way how you interpret the results of your statistical analysis in terms of causal connections (line 95) in parts of the paper. In particular, statistical co-occurrence or increased conditional probabilities do not imply any "interaction" or causal relationship; but this is how you seem to interpret (some of) your results, for instance you use terms like "interaction" (lines 104 and 439ff), "played a role in…." (line 458), "relevance of…. for …" (line 101), "dynamical driver" (line 18, line 35), "have an effect on …" (line 181, line 207), and numerous other occurrences.*

Thank you for pointing out this issue. It was not our intention to imply causation based on statistical correlation analyses only. Rather, we use several lines of evidence to infer or suggest causation. These lines of evidence include processes identified and described in the published literature summarized in the introduction and the discussion section, the results of case study analyses (not all of them included in the current manuscript), and our statistical analyses (explained in more detail in the reply to next comment). However, this has not come out very clearly yet from our text and there are indeed some misleading formulations, and we will change the text accordingly.

*As a consequence, I do not agree with your conclusion on line 469ff ("relevant for…", "play a role…"). To be sure, you have shown that during RRWP episodes there is a larger probability of heat wave occurrence. However, this does not imply that "RRWPs increase the duration of …. hot spells" (line 469/470). The latter formulation suggests that the existence of RRWPs makes an active contribution towards the occurrence of a hot spell. But it could be just as well the other way around: episodes of persistent hot spells (associated with quasi-stationary large-amplitude Rossby waves) may lead to your diagnostic of RRWPs indicating large values.*

*Let me explain. One key question that you address in this paper is: "Are heatwaves more likely to occur during periods of RRWPs or QRA?" We know that both RRWPs and QRA have a strong association with large Rossby wave amplitudes (in the case of QRA this results from what I said above), and large-amplitude Rossby waves are known to increase the likelihood for heatwaves in summer (Fragkoulidis and Wirth, 2018, and several other papers). From this perspective it appears fairly natural to expect that the likelihood for heat waves over Australia increases in case of RRWP or QRA conditions, and the question does not appear to be very interesting, or (to put it more scientifically) your hypothesis is not very daring.*

Indeed, there is robust case study, climatological and theoretical evidence that large-amplitude RWPs do increase the odds of near-surface temperature extremes. However, at least for some impacts it is not only the "simple occurrence" of an extreme (however one defines the term) that matters, but very often it is the duration of the event that is important too. Exactly this aspect (the duration) of heatwaves can be understood with the aid of the Weibull-regression analysis as this analysis quantifies by how much each quantile of the spell duration distribution shifts per unit increase in any covariate (R in this study). Again, note that the Weibull-regression analysis ultimately only diagnoses "co-variability" of large R and long-duration hot spells and not causation. However, given the aforementioned physical understanding of how individual RWPs affect near-surface temperature we propose that the duration of hot spells is enhanced by the recurrence of the RWPs.

Also, please note that there are two separate statistical analyses and apparently the text was not clear which conclusions are drawn from which analysis. The first analysis links hot spell durations with RRWPs statistically using Weibull regression, while the second is a co-occurrence analysis of RRWPs, and QRA for SEA heatwaves. We will revise the main text and the abstract to make it more clear.

Fragkoulidis and Wirth (2018) analyze large-amplitude Rossby waves. Here, we analyze "recurrent" Rossby waves, which is a subset of amplified Rossby waves where the amplified

waves recur in the same phase on a subseasonal timescale. We will clarify this point in the introduction (including the formulation of the research questions) and in the abstract, and we will include Fragkoulidis and Wirth (2018) in the literature overview. The recurrence is important for the persistence of the heatwave, in the case of Australia it ensures the recurrent formation over South-eastern Australia and hence upper-level conditions favorable for heatwave formation. show that the SEA heatwaves were composed of fast-moving Rossby waves recurring in the same phase as opposed to slow-moving Rossby waves which have been studied in the context of Northern Hemisphere heatwaves. We would further argue that more emphasis has been put on Northern Hemisphere heatwaves in recent years and thus, this work is an important study in the context of Southern Hemisphere heatwaves.

On a general level, we find the question of "how long-duration hot spells come about" is a relevant and valid research question and the presented Weibull-analysis goes beyond previous studies in this regard. Moreover, the statistical relationship is also not loose, as can be inferred from the rather rigorous significance testing, we performed.

**Reply to reviewer 2**

We thank the reviewer for taking time to evaluate this paper and appreciate their constructive remarks. We have responded to the points inline in blue font.

*Synopsis: Ali et al. analyse the relation of recurrent Rossby wave packets (RRWPs), quasi-resonant amplification (QRA), and atmospheric blocks in the Southern Hemisphere with an emphasis on southeastern Australian heat waves. Two case studies for the prominent heat waves in 2004 and 2009 motivate a climatological analysis. The authors find a significant relation between RRWPs and QRA and demonstrate that heat waves are two times more likely during QRA conditions. Overall, the study is well written and the results are clearly presented. However, I have three concerns which need to be addressed. Once these have been addressed in a suitable manner I am very happy to provide a detailed review of the manuscript.*

*Major:*

*1) According to the QRA concept, resonance can lead to high-amplitude quasi-stationary planetary waves if the combined orographic and thermal forcing pattern is sufficiently large. For the Southern Hemisphere, I assume the orographic forcing term to be of smaller magnitude than in the Northern Hemisphere and thus the thermal forcing term to dominate over the orographic forcing term. Accordingly, my interpretation is that the thermal forcing, which includes the zonal gradient of the azonal temperature at 300 hPa, becomes particularly strong during heat waves. The interpretation of the thermal forcing is therefore tricky as it could in principle be a result of the temperature anomalies rather than a forcing for large wave amplitudes. For these reasons I have doubts that the QRA concept is easily applicable during such periods and care must be taken when interpreting the results. At least a detailed discussion of this issue and how one can distinguish between forcing and results of the forcing should be included in the manuscript.*

As often, when discussing the interaction of temperature patterns and corresponding atmospheric circulation cause and effect are not strictly separable in this case. Next to theoretical considerations (see e.g., Petoukhov et al. 2013), idealized model experiments would be necessary to clearly separate cause and effect under these conditions. According to Petoukhov et al. (2013) a minimum amount of effective forcing is necessary, which is implemented as an above average forcing (amplitude within the highest 60%) in Kornhuber et al.

While we acknowledge the potential effect of surface temperature fields on upper-level temperature fields during events investigated (that might be highly barotropic), we find that in the Southern hemisphere, for wave-5 for example, the waveguide condition is filtering out most days, while the number of events changes by 15% only from 151 to 129 when applying the forcing condition that relies on the 300 mb azonal temperature fields. (see Tab. 2 in Kornhuber et al. 2017 J.Clim). We will add this point in the discussion section of the paper.

*2) I very much appreciate that the authors decided to include two case studies. These nicely illustrate the approach that is later used from a climatological perspective. However, to my impression the discussion of the cases is not very goal oriented. This is perhaps also reflected in*

*the abstract which does not include a clear outcome of the case study analyses. For example, SST anomalies are shown (Figs. 6f, 8f) but their relation to the RRWPs, blocking, or QRA is not discussed at all. If the purpose of the case studies is to explain the methodology used later on, my suggestion is to only present one case study. If the purpose is to highlight certain dynamical processes, I think the discussions in Section 3.2.1 and 3.2.2 need to be revised in a sense that the main outcomes are immediately clear to the reader.*

Thank you for this suggestion. We will modify the heatwaves case studies with a clearer focus on RRWPs to bring out our main message and remove SST composites from the figures. We will also correct labeling mistakes with blocks and Rossby waves trains in the case study descriptions and the accompanying figures. However, we prefer to stick with the two cases as the two cases show some variability: e.g., both cases have amplified Rossby waves but one of them does not show up in the QRA-metric. The discussion section will also be modified accordingly. We will also make changes to the abstract to bring out the goal of the study more clearly.

*3) One main conclusion is that RRWPs exhibit a significant relation to the duration of heat waves. To my understanding this result is plausible since southeastern Australian heat extremes, for example, occur in a highly amplified flow. Accordingly, prolonged periods of high amplitude Rossby waves favour the occurrence of several heat extremes which then are identified as one long-lasting heat wave. What is less obvious but probably at least equally important is the relation of RRWPS and the heat wave magnitude. A possible scenario would be that a first upper-level ridge leads to a heat extreme which dries the soil and thus favours higher temperatures later on through enhanced sensible heating. Could the authors therefore comment on the relation of RRWPs and heat wave magnitude, and how this differs from "ordinary non-recurrent RWPs"? Such an analysis would be extremely insightful in an operational context.*

The reviewer raises an interesting hypothesis about the temporally varying formation processes of heatwaves and suggests that, through soil desiccation, the long duration of heatwaves associated with RRWPs might also lead to anomalously strong heatwaves. Physically, the suggested hypothesis is plausible, as soil desiccation is a well-known contributor to so-called mega heatwaves (Schumacher et al. 2020; Miralles et al., 2014 https://www.nature.com/articles/s41561-019-0431-6 https://www.nature.com/articles/ngeo2141). However, the processes involved in the formation of heatwaves are drivers and, besides diabatic heating (i.e., surface and turbulent heating in the boundary layer), also involve adiabatic compression in subsiding air and temperature advection. It is, therefore, not a priori clear to what extent the suggested mechanism is relevant on a global scale and/or for SEA heatwaves. Martius et al. (2021) find a contribution of the soil moisture to a heatwave of several degrees in a idealized model setting where soil moisture anomalies over Australia are kept at -1STD for an entire season (https://journals.ametsoc.org/view/journals/clim/34/22/JCLI-D-21-0130.1.xml) The Fig. 1 below shows a weak positive correlation between the R-metric and the daily-max 2-m temperature over SEA. Nevertheless, we feel that the hypothesis raised by the reviewer might warrant its own study, and we will consider performing more in-depth analyses on this topic in the future and we will add that point to the discussion of the results.

[Figure]

Fig 1: Relationship between daily-max 2 m temperature (T2M) over SEA and the corresponding R-metric values over SEA during days identified as part of heatwaves in this study. Daily max T2M values for SEA were calculated by averaging over 141E to 153E and 29S to 43S and applying a land-sea mask to remove values over sea. Red dotted line shows a linear fit with a slope of 0.23.

---

## Author Comment (AC3)

**Final response to the editor**

We thank Michael Riemer, the editor, for his constructive comments and suggestions. In the light of the editor's suggestions, we have omitted the QRA metric and the accompanying analyses from the revised version of the manuscript. The revised title reads as "Recurrent Rossby waves and south-eastern Australian heatwaves".

Another major comment raised by the editor and reviewer #1 was on the issue of correlation vs causation. Both the reviewer #1 and the editor interpret that from the "co-occurrence" or "co-variability" of R-metric with SEA heatwaves, we conclude that RRWPs increase the duration heatwaves. We noted in our previous response that there were indeed some misleading statements in the text, and we have removed those in the revised manuscript. We clarify the following points: the analysis from Weibull regression, where we evaluate the hypothesis whether increase in R-metric is statistically associated with increased duration of hot spells, shows that increase in R-metric is associated with statistically significant increase in hot spells, including over south-eastern Australia (SEA). This information is combined with process knowledge gathered in the published literature highlighting the important role of upper-level ridges and associated subsidence for the formation of heatwaves. We supplement this analysis with discussion of the two persistent and extreme heatwaves where we show how recurrent Rossby waves play a role to form recurrent ridges over SEA. Apparently, this point did not come out clearly from the manuscript for both reviewer#1 and the editor. So, we have stressed this point more in the revised manuscript. We again take up this point in the discussion. We have further included a short discussion on a potential soil moisture feedback.

Furthermore, reviewer #1 and the editor present an alternate causal pathway where the surface anomalies may be driving the amplified waves. We thank them for this suggestion as this possible pathway was missing from our discussion so far. We have now added a discussion of this point in the revised manuscript. Based on idealised simulations by Martius et al. (2021) we argue that the effect of the surface temperature anomalies on the upper-level flow and hence the R metric is very small. Martius et al. (2021) investigates the effects of soil moisture anomalies over Australia on the local and remote flow. An ensemble of 50 CESM simulations with soil moisture set to -1 and +1 STD over Australia are analysed. The soil moisture anomalies result in surface temperature anomalies of up to 4°C and can hence serve as a proxy for a heat wave (see Fig.1a below). The temperature anomalies do have a significant effect on the geopotential height at 250hPa (Fig.1b below) and the meridional wind (Fig. 2), however the absolute anomalies are small. The meridional wind anomalies are on the order of 0.5 to 1 m/s locally. Considering that the meridional wind is averaged latitudinally for the computation of the R metric, the absolute effect on the R-metric is very small. The link between the surface temperature anomalies and upper-level flow anomalies were derived using a model and not "simply" a hypsometric equation and therefore, include feedback processes between the temperature anomaly and the circulation such as changes in precipitation and therefore diabatic heating.

Fig. 1: a) adapted from Fig.2 in Martius et al. (2021). Colours show the ensemble two-meter temperature difference between the wet and the dry simulations, solid contours show the mean sea level pressure difference. b) adapted from Fig. 6 in Martius et al. (2021). Colours show the zonal wind in m/s and dashed lines the difference in geopotential height at 250hPa (5,10 m magenta, -5,-10m blue lines) between the wet and the dry simulations

---

## Referee Report (RR1)

This study employs reanalysis data to explore the role of recurrent Rossby wave packets (RRWPs) and blocks in the persistence of southeastern Australia (SEA) heat waves using a combination of statistical analyses and case studies. As is the case with any observational study of a limited set of unique extreme events, understanding and quantifying the role of an individual Earth system component (the atmospheric flow in this case) is a challenging but important research effort. With this in mind, the authors have managed to illuminate aspects of SEA heat waves and provide new evidence on the role of the upper-tropospheric circulation. I have listed below my comments on the presented material - some of which could be deemed as "major issues" - as well as a few suggestions and corrections toward a revised version of the manuscript.

**Comments/Issues**

1. Lines 96-99 are not that clear. The 1.3 PVU threshold is an upper limit, lower limit, or what? Given the multiplication with -1, anticyclonic anomalies correspond to negative PV anomalies in the SH, right? What does it mean that with a 1.0 PVU threshold there are no blocks found over SEA? Is this a stricter or softer threshold? Is this sentence used to imply that blocking detection is too sensitive to the threshold used? Why is the blocking count over SEA used as an indication for the sensitivity?

2. Figures 4, 6 are not introduced anywhere in the text. There are just references to them and the information contained in their captions. An introductory sentence about the aim of these figures would be good. Furthermore, it is worth noting in the text that these Hovmöller diagrams do not contain information of the flow right over Australia (averaging is done between 35°-65°S). It is also worth mentioning that warm air advection from the desert and semi-arid parts of the continent toward SEA can be rather significant even with weak (lower- and/or upper-level) winds. These two aspects are relevant e.g. when considering the fact that the 2004 heat wave seems to start prior to the passage of strong RWPs over the Australian longitudinal range (Fig. 4).

3. Lines 230-236, 364: The description of the synoptic evolution of these days suggests that RWPs and blocks are independent entities of the flow and that the blocks appear to "initiate" the RWPs. Is this really justified/proved by Figs. 4 and 5? The identified blocks are not really isolated features and waves of certain amplitude do exist upstream and downstream. In addition, it is claimed - I guess unintentionally - that RWP P3 is initiated by both B3 and B4.

4. Section 3.3 investigates the relation between RRWPs and SEA heatwaves. One aspect of the statistical analysis (Lines 272-275, Table C1) - that the authors do acknowledge - is that the list of high R_SEA days contains all days with strong cyclonic PV anomalies over SEA as well. Considering the fact that ridges/blocks are associated with lower R values than troughs (Lines 405-410, Figure D1), it is plausible that the days exceeding the 90th percentile in SEA R are predominantly associated with cyclonic PV anomalies. This creates an undesirable bias in the high R_SEA sample, that can easily be eliminated in my opinion. The authors could just discard the DJF days when the average PV anomaly over SEA is above 0 (i.e., cyclonic), that is 50% of the days, and calculate the 90th percentile in R based on the remaining (ridge) days.

We will then have a more homogeneous sample of 176 high R_SEA days, X% of which co-occur with SEA HD. It would be interesting to see the new results in the last row of Table C1. What is the HD frequency increase when the ridge over SEA is associated with an RRWP rather than an individual RWP?

5. Lines 286-292: The lack of preferred PV anomaly phase on days that do not feature a SEA heatwave is not that surprising. What is a bit strange is the predominantly negative PV anomalies throughout the hemisphere on these days (Fig. 8b). What causes this? Is it perhaps because years 2011-2018 do not contribute to the mean climatology and PV anomalies in these warmer years are standardized based on a "cooler" DJF distribution?

6. Figure 9: First, the contour values of the kernel density are missing. The preferred phasing in SEA heatwave days is clear and, as mentioned before, not too surprising. It is interesting though that this only occurs in wavenumber 4. Besides, what can be said about the wavenumber 4 amplitude between HD and non-HD? It seems that the mean distance from the complex plane origin (which, if I'm not mistaken, corresponds to the amplitude) is similar in the 2 sets of days. Is there perhaps an increase in the amplitude of larger wavenumbers during SEA heatwave days?

7. Appendix A: It is indeed interesting that Southern Hemisphere R is higher in summer than winter. I wonder whether this is associated with the fact that SH storm tracks are spiralling in winter but remain rather circular in summer (e.g., Hoskins and Hodges 2005; https://doi.org/10.1175/JCLI3570.1). In any case, the fact that SH summer provides a favorable stage for RRWPs is an aspect worth mentioning in a more prominent part of the text. On a technical note, it is not clear how are the R anomalies in Fig. A1 computed and how can we compare the typical R values in the two hemispheres, if the two Hovmöller diagrams (most probably) refer to different mean climatologies.

**Minor issues**
1. Line 17: ERA-I is also an observation-based dataset. Use instead, e.g., "weather station observations" or similar.
2. Line 30: "extratropics"
3. Line 45: "part of a synoptic-scale"
4. Line 53: "the persistence of"
5. Line 56: This sentence is not really contradicting the previous. So, "however" is not fitting here.
6. Lines 66-67: Is SST and horizontal (I suppose "zonal" was meant here) velocity still used in the revised version of the paper? In addition, Figs. 4, 6, and B1 make use of a "daily maximum 2m temperature". Is this another field from ERA-I that should be mentioned here, or estimated somehow from the 6-hourly 2m temperature?
7. Line 69: Is this reference period (1980-2010) also used for the blocking feature detection?
8. Line 71: "used to quantify the recurrence"
9. Line 90: "the wave packet envelope"

10. Line 91 and elsewhere in the text: "complex plane" (not plain)
11. Line 92: Specify the section in which this phase-amplitude distribution is used.
12. Line 94: "between the 500 hPa and"
13. Line 102: "QRA conditions" should be removed
14. Line 106: What does "high-quality" mean? Has there been a study that evaluates the quality of BoM's monitoring network against others?
15. Line 108: 2019 is not used in the other fields.
16. Line 111: "were on average"
17. Line 114: "a day that is part", "SEA heatwave day (HD)"
18. Line 116: "averaged between 130°E and 153°E, which corresponds to the SEA longitudinal range" ...since "over SEA" is not correct (R is computed over a latitudinal band that lies to the south of Australia).
19. Line 117: "A sensitivity test"
20. Line 124: "1-degree horizontal resolution"
21. Line 133: "Higher numbers of"
22. Line 161: "rejecting the null"
23. Lines 165-171: All the information here is also included in the previous paragraph.
24. Line 232: "(B3 in Fig. 5d)"
25. Line 243: "windy conditions fueled many catastrophic fires"
26. Line 247: "Several RWPs"
27. Line 248: "The RWPs prior to"
28. Line 259: "moving block" sounds strange. Moving ridge perhaps?
29. Line 272: A verb would make this sentence more formal.
30. Line 276: "explore why some"
31. Line 328: "presents the relationship"
32. Line 338: "zonal wavenumber of meridional wind in the complex"
33. Figure D1: The x-axis label should be corrected ("pseudo"). In addition, the time axis direction could be the same as in the other Hovmöller diagrams of the study for consistency. In addition, the domain limits in the caption are different from the ones mentioned in Lines 401-402.
34. Lines 406, 411: The figure references need correction (D1 instead of C1)
35. Line 411: "where DJF blocks show R"

---

## Author Response (AR2)

**Author Replies**

We are grateful to both reviewers for the time they took to review and for their thoughtful comments which helped to improve this work. We have replied inline to the comments.

**Reviewer 1**

This study employs reanalysis data to explore the role of recurrent Rossby wave packets (RRWPs) and blocks in the persistence of southeastern Australia (SEA) heat waves using a combination of statistical analyses and case studies. As is the case with any observational study of a limited set of unique extreme events, understanding and quantifying the role of an individual Earth system component (the atmospheric flow in this case) is a challenging but important research effort. With this in mind, the authors have managed to illuminate aspects of SEA heat waves and provide new evidence on the role of the upper-tropospheric circulation. I have listed below my comments on the presented material - some of which could be deemed as "major issues" - as well as a few suggestions and corrections toward a revised version of the manuscript.

**Comments/Issues**
1. Lines 96-99 are not that clear. The 1.3 PVU threshold is an upper limit, lower limit, or what? Given the multiplication with -1, anticyclonic anomalies correspond to negative PV anomalies in the SH, right? What does it mean that with a 1.0 PVU threshold there are no blocks found over SEA? Is this a stricter or softer threshold? Is this sentence used to imply that blocking detection is too sensitive to the threshold used? Why is the blocking count over SEA used as an indication for the sensitivity?

The -1.0 PVU threshold in vertically averaged PV fields is a less stringent threshold than -1.3 PVU. We have added more information in the text to make this clearer.
The blocking over SEA is used to check whether the blocking algorithm, with a less stringent threshold, can identify the ridges over SEA as blocks or not for the two case studies.

Please note that the PV fields are multiplied with -1 for the rest of the analysis except for calculating blocks. This is also updated in the Method section as below:

"*PV fields are used as it is for calculating the blocking fields. However, for rest of the analysis, the PV fields are multiplied by minus one, which implies that negative (positive) PV anomalies represent anticyclones (cyclones) similar to the NH.* "

We have also added more details on the blocking algorithm:

*"Atmospheric blocking data is computed following the methodology of Schwierz et al. (2004) as in Rohrer et al. (2020) and Lenggenhager and Martius (2019). The detection scheme identifies persistent anticyclonic PV anomalies vertically averaged (VAPV) between 500 hPa and 150 hPa vertical levels. First, the VAPV anomaly is computed from the 30-day running mean climatology of the corresponding time step of the year for the years 1979–2018. An additional 2-day running mean filter is applied to smooth out high frequency transients. Then the algorithm identifies areas with VAPV ≥ 1.3 PVU in the SH. The identified areas having a persistence criterion of 5 days, and a minimum overlap of 0.7 between consecutive time steps are classified as blocks. Blocking fields identified with this algorithm are available at 6 hourly temporal resolution and 1° × 1° spatial resolution. We tested the blocking fields with a less stringent threshold of VAPV ≥ 1.0 PVU for the two case studies and did not find blocking directly over SEA. The code used to calculate blocks is available on GitHub (See Code and data availability)."*

2. Figures 4, 6 are not introduced anywhere in the text. There are just references to them and the information contained in their captions. An introductory sentence about the aim of these figures would be good. Furthermore, it is worth noting in the text that these Hovmöller diagrams do not contain information of the flow right over Australia (averaging is done between 35°-65°S). It is also worth mentioning that warm air advection from the desert and semi-arid parts of the continent toward SEA can be rather significant even with weak (lower- and/or upper-level) winds. These two aspects are relevant e.g. when considering the fact that the 2004 heat wave seems to start prior to the passage of strong RWPs over the Australian longitudinal range (Fig. 4).

As per the reviewer's suggestion, we have added an introductory paragraph to introduce the two figures. We have also mentioned now that the latitudinal bands for the Hovmöller diagram are chosen such that to capture the jet and the synoptic-scale Rossby waves. The 35°S to 65°S is an appropriate choice consistent with Röthlisberger et al. (2019).

*"Figure 4 shows the flow conditions prior to and during the heatwave and the corresponding T2m anomalies over SEA. The Hovmöller diagram (Fig. 4b) uses the 35° S – 65° S latitudinal-band for averaging V250 fields. The choice of the latitudinal band is motivated by the position of the jet suited to identify RWPs. Figure 5 shows the upper-level flow at different time steps prior to and during the heatwave. We used the two figures (Fig. 4 and Fig. 5) to study the role of transient RWPs and blocks during the heatwaves and present that next."*

We have highlighted the role of warm air advection from the continental Australia in the lines 212–214 and added the sentence suggested by the reviewer as shown below:

*"The surface flow associated with anticyclonic anomalies may also advect warm continental air due to the north westerly flow at lower levels (e.g., Parker et. al. 2014b). The warm advection associated with the surface flow can be significant even with weak upper or level winds. "*

On the reviewers point of 2004 heatwave starting prior to the passage of strong RWPs: Yes, we agree that even with weak upper-level flow there may be significant warm advection at lower level, contributing to the heatwave. However, for this case specifically, we observe an upper-level wave breaking over SEA around 4 February (Fig. 1), which could have helped the lower-level advection (Parker et. al. 2014b). Whereas, in the presentation of the results, we have only focused on the role of RRWPs rather than discussing all the possible factors responsible for the heatwave.

[Figure]

*Figure 1: Synoptic chart at 2004-02-04 12:00 (a) shows meridional wind at 250 hPa and (b) shows standardized 2-m Temperature (T2M) anomalies of daily maximum T2M. Hatches in (a) and (b) show blocks and grey-coloured solid lines show 2 PVU contour at 340 and 350 K isentropes.*

3.Lines 230-236, 364: The description of the synoptic evolution of these days suggests that RWPs and blocks are independent entities of the flow and that the blocks appear to "initiate" the RWPs. Is this really justified/proved by Figs. 4 and 5? The identified blocks are not really isolated features and waves of certain amplitude do exist upstream and downstream. In addition, it is claimed - I guess unintentionally - that RWP P3 is initiated by both B3 and B4.

Thank you for pointing it out. It is not our intention to say that RRWPs and blocks are independent entities of the flow, rather the opposite that the two features are closely related. RRWPs (i.e., eddies) upstream of a block can sustain the block (e.g., Shutts 1983; Hoskins et al. 1983). And RRWPs can form downstream of blocks because of the near constant phase of the wavebreaking (trough) on the downstream flank of blocks (Röthlisberger et al., 2018; Barton et al., 2016). Indeed, Barton et al., (2016) and Röthlisberger et al., (2018) show that blocks can initiate RWP downstream. We have now mentioned it in lines 40–45. However, here, we agree with the Reviewer's comment that the cause and effect for the block B3 initiating P3 and P4 is difficult to conclude just by Figs 4 and 5. Therefore, we have modified the text as (Lines 247–248):

*"Another set of RWPs (P3 and P4 in Fig. 4b) are associated with a block over the Pacific Ocean (B3 in Fig. 4b, 5d)"*

Thank you for spotting the mistake of RWP P3, we have corrected that.

4.Section 3.3 investigates the relation between RRWPs and SEA heatwaves. One aspect of the statistical analysis (Lines 272-275, Table C1) - that the authors do acknowledge - is that the list of high R_SEA days contains all days with strong cyclonic PV anomalies over SEA as well. Considering the fact that ridges/blocks are associated with lower R values than troughs (Lines 405-410, Figure D1), it is plausible that the days exceeding the 90th percentile in SEA R are predominantly associated with cyclonic PV anomalies. This creates an undesirable bias in the high R_SEA sample, that can easily be eliminated in my opinion. The authors could just discard the DJF days when the average PV anomaly over SEA is above 0 (i.e., cyclonic), that is 50% of the days and calculate the 90th percentile in R based on the remaining (ridge) days. We will then have a more homogeneous sample of 176 high R_SEA days, X% of which co-occur with SEA HD. It would be interesting to see the new results in the last row of Table C1. What is the HD frequency increase when the ridge over SEA is associated with an RRWP rather than an individual RWP?

The reviewer mentions that ridges/blocks are associated with lower R values than troughs citing lines 405-410 (in the previous submission) and Figure D1; however, Figure D1 (now Fig. E1) shows high $R$ values both upstream and downstream of the centroid of the block. Even at the centroid of the block, the $R$ values are higher than the background (i.e., away from the block).

The reviewer suggested to filter out all the days in DJF containing a cyclonic PV anomaly over SEA. This filters out almost 50% of the days in DJF leaving 1641 days. The high $R$ days computed from this sample gives 164 days, 57 of which co-occur with SEA HD. This increases the conditional probability to 0.34 as the reviewer hypothesized. This result has been added to Table C1 and also stated in lines 290–294. We thank the reviewer for this suggestion. However, we would like to clarify that we have not optimized the high $R$ threshold chosen for the highest co-occurrence ratio with SEA heatwaves as this is not the aim of the study

On the last question, *what is the HD frequency increase when the ridge over SEA is associated with an RRWP rather than individual RWP?*

This is an intriguing question. However, it is quite challenging to define an individual RWP. Here, we use the RWP dataset from Frougdoulis et al. (2020) and are grateful to them for sharing their dataset. They calculate a 2-D RWP amplitude for each time step. Hence, to compare the RWP with R-metric, we average the RWP data between 30S–65S and then average it over SEA longitude, let's call this E. Next, we categorize high E days representing amplified RWP activity over SEA by taking 90th percentile of E for DJF, same way as we define high R days over SEA. Please note that this is a quick preliminary analysis as to we did not check which threshold is best for RWP amplitude. Therefore, the results below should be carefully interpreted are not optimized to the best hit rate for each metric.

We obtain 352 days with high E of which 40 overlap with SEA HD. Out of these 40 days, only 8 days include high *R* days. Whereas 67 high *R* days overlap with SEA HD. In the DJF climatology, we find that 63 days categorized as high E days overlap with high R days. It is possible to have high *R* day but not high E day as the high R days have an additional condition for recurrence in the same phase.

*Fragkoulidis, G., & Wirth, V. (2020). Local Rossby Wave Packet Amplitude, Phase Speed, and Group Velocity: Seasonal Variability and Their Role in Temperature Extremes, Journal of Climate, 33(20), 8767-8787.*
*https://journals.ametsoc.org/view/journals/clim/33/20/jcliD190377.xml*

5.Lines 286-292: The lack of preferred PV anomaly phase on days that do not feature a SEA heatwave is not that surprising. What is a bit strange is the predominantly negative PV anomalies throughout the hemisphere on these days (Fig. 8b). What causes this? Is it perhaps because years 2011-2018 do not contribute to the mean climatology and PV anomalies in these warmer years are standardized based on a "cooler" DJF distribution?

The PV climatology is calculated using 1979–2018 as mentioned in the caption of Figure 8. We have removed the climatology period in the Method to avoid confusion (L69 in previous version). We found a mistake in the code when calculating PV anomalies. The Figs. 8a and 8b have been updated and we do not see predominantly negative anomalies in Fig 8b as before. The Section 3.3 is updated as per the changes in Figs. 8 and 9.

6.Figure 9: First, the contour values of the kernel density are missing. The preferred phasing in SEA heatwave days is clear and, as mentioned before, not too surprising. It is interesting though that this only occurs in wavenumber 4. Besides, what can be said about the wavenumber 4 amplitude between HD and non-HD? It seems that the mean distance from the complex plane origin (which, if I'm not mistaken, corresponds to the amplitude) is similar in the 2 sets of days. Is there perhaps an

increase in the amplitude of larger wavenumbers during SEA heatwave days?

We have updated Fig 9 with contour values and added distribution for the climatology as well. We have additionally added wavenumber 5 because we also see a wavenumber 5 pattern in the updated Fig 8a. The WN4 and WN5 amplitude (distance from the origin) is expected to be higher than climatology for high R days. This is also what we observed in the Fig. 8 (a, b, d, e).

[Figure]

*Figure 2. Bivariate kernel density estimate using Gaussian kernels in the complex plane of the Fourier decomposed meridional wind at 250 hPa averaged between 35°S and 65°S. Only zonal WN4 (top) and WN5 (bottom) are shown for days belonging to (a, d) high RSEA and SEA HD, (b, e) high RSEA and non-SEA HD, and (c, f) DJF climatology.*

As mentioned in the updated text, the wavenumbers shown in Figure 9 are motivated from the WN4 and WN5 patterns observed in Fig 8a, and the difference in phase and amplitude distribution of other wavenumbers shown below (Fig. 3) is relatively smaller.

[Figure]

*Figure 3: Same as Figure 2 but for WNs, k=5, 6, 7.*

7.Appendix A: It is indeed interesting that Southern Hemisphere R is higher in summer than winter. I wonder whether this is associated with the fact that SH storm tracks are spiralling in winter but remain rather circular in summer (e.g., Hoskins and Hodges 2005; https://doi.org/10.1175/JCLI3570.1). In any case, the fact that SH summer provides a favorable stage for RRWPs is an aspect worth mentioning in a more prominent part of the text. On a technical note, it is not clear how are the R anomalies in Fig. A1 computed and how can we compare the typical R values in the two hemispheres, if the two Hovmöller diagrams (most probably) refer to different mean climatologies.

R anomalies for each day of year at each longitude are calculated from the mean of the day of the year mean. Therefore, the R anomalies at each longitude show variation with the mean of the day of year mean and have a seasonal pattern. The magnitude of the anomalies shows that there is larger variation in the values for the NH than the SH. We have updated the description Lines 410 onwards to make it clearer. As the reviewer rightly points out, R is higher in austral summer than winter.
This is indeed an interesting result. However, we did not include in the main text because:
a. More analysis is needed to explain the difference observed between the Southern and Northern Hemisphere.
b. Comparing NH R-metric with SH R-metric does not quite fit with the storyline of the paper. Hence, we have included it in the Appendix. Thank you for also sharing Hoskins and Hodges (2005) reference.

**Minor issues**
We thank the reviewer for spotting the minor issues.

1. Line 17: ERA-I is also an observation-based dataset. Use instead, e.g., "weather station observations" or similar.
Thank you, we have implemented this suggestion.

2. Line 30: "extratropics"
Thank you, we have corrected that.

3. Line 45: "part of a synoptic-scale"
Thank you, we have corrected that.

4. Line 53: "the persistence of"
Thank you, we have corrected that.

5. Line 56: This sentence is not really contradicting the previous. So, "however" is not fitting here.
Thank you, we have implemented this suggestion.

6. Lines 66-67: Is SST and horizontal (I suppose "zonal" was meant here) velocity still used in the revised version of the paper? In addition, Figs. 4, 6, and B1 make use of a "daily maximum 2m temperature". Is this another field from ERA-I that should be mentioned here, or estimated somehow from the 6-hourly 2m temperature?
Yes, daily maximum 2m temperature is derived from the 6 hourly 2m Temperature. We have removed zonal velocity updated the Data section accordingly.

7. Line 69: Is this reference period (1980-2010) also used for the blocking feature Detection?
Blocking detection uses 1980-2018 period. We have updated the description in the Methods section (Lines 107-109).

We have also clarified the period used in the anomaly calculation where necessary (e.g., caption of Figure 8).

8. Line 71: "used to quantify the recurrence"
Thank you, we have corrected that.
9. Line 90: "the wave packet envelope"
Thank you, we have corrected that.
10. Line 91 and elsewhere in the text: "complex plane" (not plain)
Thank you, we have corrected that.
11. Line 92: Specify the section in which this phase-amplitude distribution is used.
Thank you, we have corrected that.
12. Line 94: "between the 500 hPa and"
Thank you, we have corrected that.
13. Line 102: "QRA conditions" should be removed
Thank you, we have corrected that.

14. Line 106: What does "high-quality" mean? Has there been a study that evaluates the quality of BoM's monitoring network against others?

The weather stations in this dataset are selected for their quality and length of the available temperature data. From the website of the ACORN-SAT dataset, "*The Bureau's methods have been extensively peer-reviewed and found to be among the best in the world. This is crucial, as it means the community can have confidence the Bureau is providing an accurate estimate of Australia's true temperature trend.* "

More details about the quality checks are provided here:
http://www.bom.gov.au/research/publications/researchreports/BRR-032.pdf

15. Line 108: 2019 is not used in the other fields.
Thank you for spotting this. We initially used ACORN-SAT till 2019 but later used it only until 2018 due to the availability of other datasets. We have corrected the time period as 1979–2018.

16. Line 111: "were on average"
Thank you, we have corrected that.

17. Line 114: "a day that is part", "SEA heatwave day (HD)"
Thank you, we have corrected that.

18. Line 116: "averaged between 130°E and 153°E, which corresponds to the SEA longitudinal range" ...since "over SEA" is not correct (R is computed over a latitudinal band that lies to the south of Australia).
Thank you, we have corrected that.

19. Line 117: "A sensitivity test"
Thank you, we have corrected that.

20. Line 124: "1-degree horizontal resolution"
We think that it is also correct to use "spatial resolution".

21. Line 133: "Higher numbers of"
We have modified the sentence as:
*"Over land, many hot spells are seen over parts of SEA, South Africa, and South America, having 350 or more spells."*

22. Line 161: "rejecting the null"
Thank you, we have corrected that.

23. Lines 165-171: All the information here is also included in the previous paragraph.
Yes, we have re-structure this part.

24. Line 232: "(B3 in Fig. 5d)"
Thank you, we have corrected that.

25. Line 243: "windy conditions fueled many catastrophic fires"
Thank you, we have corrected that.

26. Line 247: "Several RWPs"
Thank you, we have corrected that.

27. Line 248: "The RWPs prior to"
Thank you, we have corrected that.

28. Line 259: "moving block" sounds strange. Moving ridge perhaps?
Removed "moving". We haven't used "ridge" because it is classified as a block by the detection algorithm, a ridge would be more general.

29. Line 272: A verb would make this sentence more formal.
We have modified the sentence as the following:
*"First, we note the co-occurrence of high $R_{SEA}$ days and SEA heatwave days (SEA HD) as defined in section 2.3."*

30. Line 276: "explore why some"
Thank you, we have corrected that.

31. Line 328: "presents the relationship"
Thank you, we have corrected that.

32. Line 338: "zonal wavenumber of meridional wind in the complex"
Thank you, we have corrected that.

33. Figure D1: The x-axis label should be corrected ("pseudo"). In addition, the time axis direction could be the same as in the other Hovmöller diagrams of the study for consistency. In addition, the domain limits in the caption are different from the ones mentioned in Lines 401-402.
Thank you, we have corrected the spelling.

34. Lines 406, 411: The figure references need correction (D1 instead of C1)
Thank you, we have corrected that.

35. Line 411: "where DJF blocks show R"
Thank you, we have corrected that.

**Reviewer 2**

The authors have addressed my comments of a previous review in a suitable manner. By removing the QRA analysis, a critical aspect that would have required further consideration and analysis is taken out of the manuscript. Nevertheless, the manuscript is an important step towards understanding the atmospheric precursors of heat waves in southeastern Australia. Numerous minor inaccuracies have made it difficult to read. Once these are corrected, I recommend the manuscript for publication in WCD.

**Major comment:**
1) The R metric is calculated at the 250 hPa level. The 350-K isentropic level which is commonly used to diagnose upper-level anticyclones associated with heat waves lies at a higher isobaric level (approx 200 hPa climatological DJF mean). If you determined the R metric at the 350-K isentropic level instead, would you then find an even stronger link between high R_SEA days and SEA HDs? This would be somewhat expected as the high tropopause in such

case would correspond to high thickness and accordingly to a higher troposphere-mean temperature.

Figure 4 below shows R-metric at 200 hPa compared with 250 hPa for January 2004. The major patterns are captured well at both the pressure levels. There are only minor changes in magnitude and those changes will affect the whole distribution of R-metric. Thus, we do not expect a substantial difference in the high R days as they were selected based on a percentile from that distribution.

[Figure]

Figure 4: Comparison of R-metric at two pressure levels: 200 hPa and 250 hPa

We think that both 200 and 250 hPa levels are well suited to capture the jet. The figure 5 showing zonal mean zonal wind for DJF climatology from the ERA-40 Atlas also supports our point. Hence, we stick with 250 hPa as also used in Röthlisberger et al. (2019).

[Figure]

Figure 5. Zonal mean zonal wind for DJF climatology from the ERA-40 Atlas. Source:
https://sites.ecmwf.int/era/40-atlas/images/full/D25_XS_SON.gif

**Minor comments:**

l. 32/33: Here the authors use the terms "Rossby wave patterns" and "recurrent Rossby wave patterns". Are these any different from Rossby wave packets? If not, please consider to use consistent terminology.

Thank you for this suggestion. Here, both imply the same. We have changed it to Rossby wave packets instead of patterns and did so in the other instances.

l. 36: Perhaps remove "short" as it is a relative term or replace it with "on medium-range to sub-seasonal time scales"?

Thank you for this suggestion. We have modified it to be more specific about the time scale:
*"RRWPs can be considered as a subset of amplified Rossby waves with a condition that the transient eddies recur spatially in the same phase on a short time scale of days to weeks."*

l. 37: See comment on lines 32/33.

Please see reply to Minor point 1.

l. 45: "Rossby wave packets" instead of "Rossby wave packet"?

Thanks for the suggestion. We have modified the sense as: "These anticyclonic PV anomalies can form as part of a synoptic-scale Rossby wave packet (RWP)."

l. 47: "RWPs...break over SEA as anticyclonic equatorward (LC1-type) Rossby wave breaking" is awkward as "breaking" is redundant. Perhaps simply write "...and eventually break anticyclonically over SEA".
Thank you, we have corrected that.

l. 65: I assume you are using 6-hourly data!? Perhaps provide this information so that it becomes immediately clear later on why you are using 14.25 day running means later on.
Yes, thank you, we have added that.

l. 69: Please clarify, are you considering seasonal means, running means or something different?
We have clarified the climatology calculation where used, e.g., caption of Fig. 8 which uses seasonal means and in the Method section for T2M anomalies.

l. 70: I guess it has to be "2.2 Recurrent Rossby Waves".
Thank you, we have corrected that.

l. 99: So I guess with both thresholds 1.3 PVU and 1.0 PVU you did not find blocking directly over SEA? Please write this explicitly in the text.
Thanks for this suggestion. We have mentioned it explicitly in lines 112–113:
*"We tested the blocking fields with a less stringent threshold of VAPV ≥ 1.0 PVU for the two case studies and did not find blocking directly over SEA."*

l. 102: I guess this needs to be updated as QRA is not included any more in the manuscript.
Thank you, we have corrected that.

l. 115: I assume "HD" refers to heat day? Please introduce the acronym.
Thank you, we have corrected that.

l. 117: For TMAX, the 90th percentile is calculated for each station for each month in DJF. So, the 90th percentile for R is not entirely consistent as this is defined based on the daily mean, right? Please clarify.
Yes, the TMAX scheme is on the lines of one used in Quinting and Reeder (2017). The 90th percentile of R, based on the daily mean R, aims to identify days with R towards the extreme end of the distribution and hence serves our purpose of comparing RRWP conditions during SEA heatwaves.

l. 131: Better write "hot surface weather"? This would be consistent with the terminology "hot spell".
Agreed, thanks for the suggestion, we have changed it.

l. 147: The value of R is averaged from 35 to 65°S. The hot spells, however, are defined between 20 to 70°S. How do you treat hot spell grid points, e.g., between 20 to 35°S that are remote from the region over which R is averaged?

Although the metric R is computed from 35°S to 65°S meridional wind at 250 hPa, it only has longitude information as shown in lines 154–160.
Hence, the value of covariate R will be the same for all the spells at the same longitude. And based on that, we did not find substantial significant areas North of 30°S, implying that R does not play a significant role for most of the hot spells North of 30°S. However, the ones we get significant around the 30°S band, e.g., over SEA can be explained by the process discussed in the case studies i.e., RRWPs helping to form recurrent ridges over SEA.

Furthermore, we also checked sensitivity of the R-metric to latitudinal averaging by using 20°S to 80° S meridional wind vs 35° S to 65° S (Figure 5). We did not find differences in the seasonal anomalies of R between the two.

[Figure]

*Figure 5: Day of year R anomalies computed for each longitude and each day of year with respect to the mean of day of year means.*

I. 153: Are May and September considered at all?
No, only NDJFMA months are considered for the Weibull analysis.

I. 166-171: These lines are nearly the same as in the previous paragraph and can presumably be removed from the manuscript.
Thank you, we have corrected that.

I. 193: The heading needs to be adapted as QRA is not included in the manuscript any longer.

Thank you, we have corrected that.

l. 209 and elsewhere: Please use notation m\,s^{-1} instead of m/s.
Thank you, we have changed the notation.

l. 226: R is already at high levels from 7-12 February (grey contours in Fig. 4b). However, the shading does not indicate high values of the meridional wind. Is this due to the averaging between 35-65°N where positive and negative values in v may cancel out? A short explanation would help to avoid a potential confusion.

Commen
averaging
12 Febru
meridiona
"distracto
250°E.

This is due to the approximately 15-day running mean filter that the R-metric uses. We can see the small wind velocity pulses around 8–9 February are in-phase with the amplified wave packet P1 later (around 12 February).

l. 228: I guess you mean the labeling in Fig. 4b!? Only B1 but not P1 is labelled in Fig. 5b.
Thanks for spotting, it is corrected now.

l. 228: 2x downstream: Consider to specify the location of the block as "central Pacific" instead of "downstream of Australia" which is quite unspecific.

l. 229: The block in Fig. 5a south of Australia is this the same block as B1 in Fig. 5b? If so, please label the block also in Fig. 5a.
Yes, it is the same block, thanks for spotting it.

l. 232: The text and the labels in the Figures do not match. There is no B3 in Fig. 5b. Please carefully revise this paragraph to ensure that the labels are correct.
Thank you for spotting, it is corrected now.

l. 233: Also here, B4 is south of South Africa but not of South America. Please correct.
Thank you for spotting it.

l. 235: I guess you mean four RWPs instead of three.
Thank you for spotting it.

l. 250: Remove "forming" before "over Australia".
Thank you for spotting it.

l. 258: I guess you mean P2, P4.
Yes, thank you.

l. 259: Is it better to use the term absorption here than injection? This would be, for example, consistent with the terminology used by Yamazaki and Itoh (2009).

Thank you, we have changed it to absorption.

I. 270: Should be "Same as in Fig. 5", I assume.
Yes, thank you.

I. 308: To my impression the PV anomalies in Fig. 8e occur unexpectedly close to the equator (especially compared to where PV anomalies are found during SEA heat waves). Do the authors have any explanation for this?

Please note that we have updated Fig. 8d, 8e to have the WN4 and WN5 components for the mean PV for SEA HD and high R days.

However, in the version of the figure 8 in the last revision to which this comment refers to, Fig. 8d and 8e showed climatological PV values and not the PV during the heatwaves. When comparing the PV climatology composite for the zonal wavenumber 4 in Fig. 8e, with Fig. 8d, the location of the zonal wavenumber 4 component matches with where the amplified ridges are located (Fig. 8d). So, we do not think that this is unexpected. Also, in the case studies, the location of the 2PVU contour at 350 K varies considerably depending on the time step.

I. 313: I assume it should be non SEA HD.
Yes, thank you.

I. 327: "which suggests" instead of "and suggests"?
Yes, thank you.

I. 361: "persistent ridges" may also be referred to as blocking, but I think this is not what you would like to say as blocks were not directly observed over SEA. Please reconsider the wording.
Yes, precisely why we did not refer to "persistent ridge" as blocking. It is also arguable in the scientific community whether a subtropical ridge should be referred to as a "block".

I. 368: Although the MJO is probably known among the WCD readership, please introduce the acronym here.
That's true, we have corrected that.

I. 373: Also as a prognostic metric? This could be relevant towards sub-seasonal forecasts of heat waves/hot spells.
Thank you for this suggestion. We have added that.

**References**
Barton, Y., Giannakaki, P., Von Waldow, H., Chevalier, C., Pfahl, S., & Martius, O: Clustering of regional-scale extreme precipitation events in southern Switzerland. Monthly Weather Review, 144(1), 347–369., 2016.

Fragkoulidis, G., & Wirth, V.: Local Rossby Wave Packet Amplitude, Phase Speed, and Group Velocity: Seasonal Variability and Their Role in Temperature Extremes, Journal of Climate, 33(20), 8767-8787, https://journals.ametsoc.org/view/journals/clim/33/20/jcliD190377.xml, 2020.

Röthlisberger, M., Martius, O., & Wernli, H.: Northern Hemisphere Rossby Wave Initiation Events on the Extratropical Jet—A Climatological Analysis, Journal of Climate, 31(2), 743–760, https://journals.ametsoc.org/view/journals/clim/31/2/jcli-d-17-0346.1.xml, 2018.